# Path Development Network with Finite-dimensional Lie Group

**Hang Lou**  *louhang39@gmail.com*
*Department of Mathematics*
*University College London*

**Siran Li**  *sl4025@nyu.edu*
*Department of Mathematics*
*Shanghai Jiao Tong University*

**Hao Ni**  *h.ni@ucl.ac.uk*
*Department of Mathematics*
*University College London*

**Reviewed on OpenReview:** *https://openreview.net/forum?id=YoWBLu74TL*

## Abstract

Signature, lying at the heart of rough path theory, is a central tool for analysing controlled differential equations driven by irregular paths. Recently it has also found extensive applications in machine learning and data science as a mathematically principled, universal feature that boosts the performance of deep learning-based models in sequential data tasks. It, nevertheless, suffers from the curse of dimensionality when paths are high-dimensional.

We propose a novel, trainable path development layer, which exploits representations of sequential data through finite-dimensional Lie groups, thus resulting in dimension reduction. Its backpropagation algorithm is designed via optimization on manifolds. Our proposed layer, analogous to recurrent neural networks (RNN), possesses an explicit, simple recurrent unit that alleviates the gradient issues.

Our layer demonstrates its strength in irregular time series modelling. Empirical results on a range of datasets show that the development layer consistently and significantly outperforms signature features on accuracy and dimensionality. The compact hybrid model (stacking one-layer LSTM with the development layer) achieves state-of-the-art against various RNN and continuous time series models. Our layer also enhances the performance of modelling dynamics constrained to Lie groups. Code is available at `https://github.com/PDevNet/DevNet.git`.

## 1 Introduction

Signature-based methods are an emerging tool for the modelling of time series data. The signature of a path, originated from rough path theory in stochastic analysis (*cf.* Lyons et al. (2007); Coutin & Qian (2002) and the many references cited therein), has shown its promise as an efficient feature representation of time series data, facilitating prediction performance when combined with suitable machine learning models, *e.g.*, deep learning and tree-based models (Xie et al. (2017); Arribas et al. (2018); Morrill et al. (2019)).

The *signature* of a path $X : [0, T] \to \mathbb{R}^d$ is the central concept in the theory of rough paths, which aims at providing rigorous mathematical tools for defining and analysing solutions to the controlled differential equations (CDE) driven by oscillatory paths rougher than semimartingales. Such solutions are of very low regularity in general, hence have remained impenetrable via classical analytical tools for CDE. The celebrated theory of regularity structure (Hairer (2014)) underpins the theoretical contribution of the rough path theory in pure mathematics. Roughly speaking, the signature of a path serves as a principled feature that offers

a top-down description of the path. Just as the monomial basis of $\mathbb{R}^d$, the signature — viewed as their noncommutative analogue — constitutes a basis for the path space. More specifically, the signature of $X$ is defined as an infinite sequence $\left(1, \mathbf{X}^{(1)}_{[0,T]}, \cdots, \mathbf{X}^{(k)}_{[0,T]}, \cdots\right)$ where, providing that the integrals are well defined,

$$\mathbf{X}^k_{[0,T]} = \int_{0 < t_1 < \cdots t_k < T} \mathrm{d}X_{t_1} \otimes \cdots \otimes \mathrm{d}X_{t_k}.$$

We refer the reader to Section 2.1 and Appendix A for the precise definition of signature of paths of bounded variation (BV-paths). See *e.g.*, Lyons (2014) and the many references cited therein for the general case of paths of finite $p$-variation; $p \geq 1$.

In applications to time series analysis, the signature is a mathematically principled feature representation, in contrast to certain deep learning-based models that are computationally expensive and difficult to interpret. The signature is universal and enjoys desirable analytic-geometric properties (*cf.* Lyons (2014); Levin et al. (2013)), hence can be used as a deterministic, pluggable layer in neural networks (Kidger et al. (2019)). Nevertheless, the signature encompasses three major challenges in practice:

- It suffers from the curse of dimensionality — dimension of the truncated signature up to the $k^{\text{th}}$ term, *i.e.*, $\sum_{i=0}^{k} d^i = \frac{d^{k+1}-1}{d-1}$, grows geometrically in the path dimension $d$.

- It is not data-adaptive, and hence appears ineffective in certain learning tasks.

- It incurs potential information loss due to finite truncation of the signature feature.

The main objective of this paper is to address the above issues. We propose a novel trainable feature representation of time series, termed as the *path development layer*, which is mathematically principled, data-adaptive, and suitable for high-dimensional time series. The theoretical foundation of the path development layer proposed in our paper is rooted in the concept of the *development* of a path (*a.k.a.* Cartan development; see, *e.g.*, Driver (1995)). It has recently been explored in theoretical studies of rough paths, especially for signature inversion (Lyons & Xu (2017)) and uniqueness of signature (Boedihardjo & Geng (2020); Hambly & Lyons (2010); Chevyrev & Lyons (2016)).

One way to introduce the development of a path $X$ on $\mathbb{R}^d$ is as follows. Consider a matrix Lie group $G$. It shall serve as the range for the development. Then let $M$ be a linear transform from $\mathbb{R}^d$ to $\mathfrak{g}$, the Lie algebra of $G$. The path development of $X$ can be viewed as a generating function of the signature:

$$M \longmapsto \sum_{k \geq 0} M^{\otimes k}\left(\mathbf{X}^k_{[0,T]}\right),$$

where $M^{\otimes k}(v_1 \otimes v_2 \otimes \cdots \otimes v_k) = M(v_1) \cdot M(v_2) \cdot \cdots \cdot M(v_k)$ with $v_i \in \mathbb{R}^d$ for $i \in \{1, \cdots, k\}$. The product $\cdot$ is the matrix multiplication.

An important remark is in order. As we are working with specific matrix Lie groups $G$ and the corresponding Lie algebras $\mathfrak{g}$, it is natural to view both $G$ and $\mathfrak{g}$ as subsets of $\mathfrak{gl}(m; \mathbb{R})$, the space of $m \times m$ real matrices, *i.e.*, the Lie algebra of the general linear group $\mathrm{GL}(m; \mathbb{R})$, which consists of invertible $m \times m$ matrices. This viewpoint significantly simplifies our theories in practice.

The preservation of favourable geometric and analytic properties makes the path development a promising feature representation of sequential data. Specifically, non-commutativity of the multiplication in $M^{\otimes k}\left(\mathbf{X}^k_{0,T}\right)$ reflects the irreversibility of the order of events. It has been established in Chevyrev & Lyons (2016) that, with $\mathfrak{g}$ suitably chosen, the path development constitutes universal and characteristic features. Moreover, in contrast to the infinite dimensionality of the path signature, the development takes values in finite-dimensional Lie groups with dimensions independent of the path dimension $d$.

Motivated by the discussions above, we construct the *path development layer*, which is the central objective of this paper. This layer transforms any sequential data $x = (x_0, \cdots, x_N) \in \mathbb{R}^{d \times (N+1)}$ to the path development

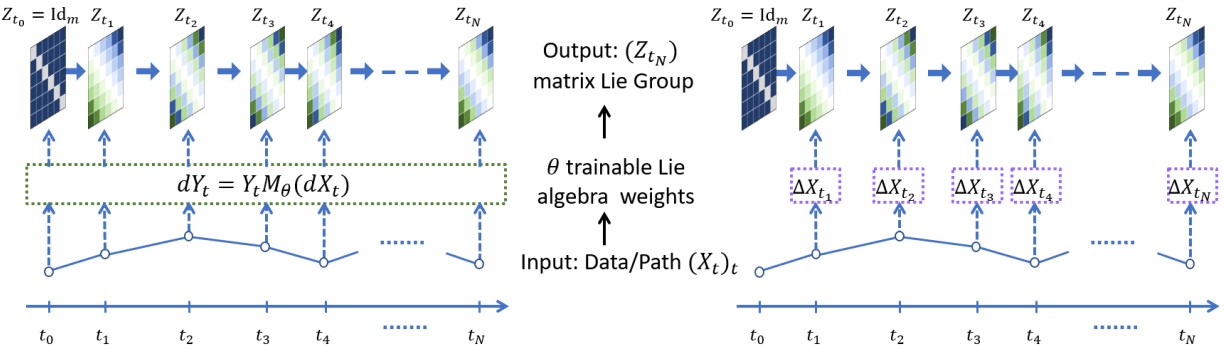

Figure 1: A high-level summary of the proposed development layer. Output and trainable weights take values in the matrix Lie group $G$ and Lie algebra $\mathfrak{g}$, respectively. (Left) It can be interpreted as a solution to the linear controlled differential equation driven by a driving path (Definition 2.2), which is a continuous lift of sequential data. (Right) It can be viewed as an analogy of the RNNs, but with a simpler form (Eq. (1)).

$z = (z_0, \cdots, z_N)$ under a trainable linear map $M_\theta : \mathbb{R}^d \to \mathfrak{g}$, where each $z_i$ lies in the Lie group $G$. For each $n \in \{0, \cdots, N\}$ we set

$$z_{n+1} := z_n \exp\left(M_\theta(x_{n+1} - x_n)\right), \quad z_0 = \mathrm{Id}_m, \tag{1}$$

where $\theta$ is the model parameter, $\mathrm{Id}_m$ is the identity matrix, and exp is the matrix exponential. It possesses a recurrence structure analogous to that of RNNs.

To optimise the model parameters of the development layer, we exploit the recurrence structure in Eq. (1) and the Lie group-valued output to design an efficient gradient-based optimisation method. We combine backpropagation through time of RNNs and "trivialisation", an optimisation method on manifolds (Lezcano-Casado (2019)). In particular, when $\mathfrak{g}$ is the Lie algebra of the orthogonal group, we can establish boundedness of the gradient. This alleviates the gradient vanishing/exploding problems of backpropagation through time, thus leading to a more stable training process.

To the best of our knowledge, this paper is the first of the kind to

1. construct a trainable layer based on the path development; and

2. design, taking into account adequate Lie group structures, the backpropagation of the development layer.

Key advantages of our proposed path development layer include the following:

1. it provides mathematically principled features that are characteristic and universal (Theorem 2.1 and Theorem B.1);

2. it is a data-adaptive, trainable layer pluggable into general neural network architectures;

3. it is applicable to high-dimensional time series;

4. it helps stabilise the training process; and

5. it can model the dynamics on non-Euclidean spaces by exploiting appropriate Lie group structure.

Numerical results reported in this paper validate the efficacy of the development layer in comparison to the signature layer and several other continuous models. Moreover, the hybrid model obtained by stacking together LSTM with the development layer consistently achieves outstanding performance with more stable training processes and less need for hyper-parameter tuning. We also provide toy examples for simulated

Brownian motions on $\mathbb{S}^2$ and $N$-body motions. They serve as evidence for the idea that *equivariance structures* inherent to learning tasks can be effectively incorporated into the development model by properly choosing Lie groups to result in performance boost. This may offer a novel, promising class of models based on development modules for time series on manifolds or trajectories of moving data clouds.

## 1.1 Related works

**Recurrent Neural Networks (RNNs)** RNNs and their variants are popular models for sequential data, which achieve superior empirical performance on various time series tasks (Hochreiter & Schmidhuber (1997); Cho et al. (2014); Bai et al. (2018)). They show excellent capacity for capturing temporal dynamic behaviour, thanks to the recurrence structure of hidden states. However, they are also prone to difficulty in capturing long-term temporal dependency and issues of vanishing/exploding gradients (Bengio et al. (1994)). Restricting weight matrices of RNNs to the unitary group $U(n)$ may circumvent such issues, as shown in Arjovsky et al. (2016); Lezcano-Casado (2019); Kiani et al. (2022). The development layer proposed in this work possesses recurrence structures similar to those of the RNNs (see Eq. (1)), but in an explicit and much simpler form. Furthermore, the outputs of our layer are in the suitably chosen Lie groups, which has positive effects on modelling long-term temporal dependency and stabilising the training processes.

**Geometric Deep Learning (GDL)**. Recent advances in GDL have gained enormous attention by extending neural networks to handle complex, non-Euclidean (*e.g.*, manifold-valued) data. See Monti et al. (2017); Bronstein et al. (2017); Cao et al. (2020). A notable example is Riemannian ResNet in Katsman et al. (2024), which extends the construction of Residual Neural Network (ResNet) to general Riemannian manifolds. Our proposed development resembles Riemannian ResNet — both employ the exponential map and exhibit recurrence structures. However, the recurrence of Riemannian ResNet lies in two consecutive layers, whereas the recurrence occurs between two consecutive times in a single development layer. Also, in contrast Riemannian ResNet, the development layer is designed specifically for times series input.

**State space models (SSMs).** State space models, originated from the approximation of linear dynamical systems, serve as a sequence layer that can be stacked for various time series tasks (Rangapuram et al. (2018)). More recently, the structured state space sequence ("S4") layers and its simplified version "S5" have been introduced by imposing new parameterisation of linear coefficient matrices in SSM and utilising low-rank approximation. The S4 and S5 lead to significant improvement on computational efficiency and state-of-the-art performance on long-range sequence modeling tasks. See Gu et al. (2021a); Smith et al. (2022). Analogous to the SSM models, the development layer proposed in our work is also originated from linear differential equations. Nonetheless, the output of our development layer takes values in matrix groups at each time, while the output of SSMs are vector-valued. The geometric structure of the matrix Lie groups prove to be crucial, from both theoretical and algorithmic perspectives, for the development layer proposed in this paper.

**Continuous time series modelling.** Continuous time series models have attracted increasing attention due to their strengths on treating irregular time series. Popular differential equation-inspired models include Neural ODEs (Haber & Ruthotto (2017); Chen et al. (2018)), Neural SDEs (Liu et al. (2019)), and Neural CDEs (controlled differential equations; Kidger et al. (2020); Morrill et al. (2021)). The model based on path signature and development we propose here, similar in spirit to the above, takes a continuous perspective — it embeds discrete time series to the path space and solves for linear CDEs driven by the path. We shall show its advantages in coping with time series which are irregularly sampled, of variable length, and/or invariant under time reparameterisation. Notably, recent work has also extended Neural ODEs to handle other data types, such as graph data Gravina et al.; Eliasof et al. (2024).

## 1.2 Organisation of the paper

The remaining parts of the paper are organised as follows: In §2 we collect some background materials on path signature, path development, and optimisation on manifolds. In §3 we propose our path development layer. Algorithms for forward/backward pass are described in detail with rigorous mathematical justification. Next, in §4, numerical experiments on sequential data imaging and dynamics on manifolds are reported. Brief

concluding remarks are given in §6. The four appendices at the end of the paper present various mathematical proofs and experimental details.

## 2 Preliminaries

### 2.1 Path Signature

We present here a self-contained, brief summary of the path signature, which can be used as a principled, efficient feature of time series data. See Lyons et al. (2007); Levin et al. (2013); Chevyrev & Kormilitzin (2016); Kidger et al. (2019) and the references cited therein.

Denote by $\mathcal{V}_1\left([0,T],\mathbb{R}^d\right)$ the space of continuous paths on $\mathbb{R}^d$ of finite length. Write $T\left((\mathbb{R}^d)\right) := \bigoplus_{k \geq 0}\left(\mathbb{R}^d\right)^{\otimes k}$ for the tensor algebra equipped with tensor product and component-wise addition. Signature takes values in $T\left((\mathbb{R}^d)\right)$, and its zeroth component is always 1.

**Definition 2.1** (Path Signature). *Let $J \subset [0,T]$ be a compact interval and $X \in \mathcal{V}_1\left([0,T],\mathbb{R}^d\right)$. The signature of $X$ over $J$ is defined as*

$$S(X)_J = \left(1, \mathbf{X}_J^1, \mathbf{X}_J^2, \cdots\right),$$

*where $\mathbf{X}_J^k = \int_{\substack{u_1 < \cdots < u_k \\ u_1,\ldots,u_k \in J}} \mathrm{d}X_{u_1} \otimes \cdots \otimes \mathrm{d}X_{u_k}$ for each $k \geq 1$ as Riemann-Stieltjes integrals.*

The truncated signature of $X$ of order $k$ is $S^k(X)_J := \left(1, \mathbf{X}_J^1, \mathbf{X}_J^2, \cdots, \mathbf{X}_J^k\right)$. Its dimension is $\sum_{i=0}^k d^i = \frac{d^{k+1}-1}{d-1}$, which grows exponentially in $k$. The signature can be regarded as a non-commutative version of the exponential map defined on the space of paths. Indeed, consider $\exp : \mathbb{R} \to \mathbb{R}$, $\exp(t) = e^t = \sum_{k=0}^\infty \frac{t^k}{k!}$; it is the unique $C^1$-solution to the linear differential equation $\mathrm{d}\exp(t) = \exp(t)\mathrm{d}t$. Analogously, the signature map $t \mapsto S(X)_{0,t}$ solves the following linear differential equation:

$$\mathrm{d}S(X)_{0,t} = S(X)_{0,t} \otimes \mathrm{d}X_t, \qquad S(X)_{0,0} = \mathbf{1} := (1, 0, 0, \ldots). \tag{2}$$

The range of signature of all paths in $\mathcal{V}_1\left([0,T],\mathbb{R}^d\right)$ is denoted as $S\left(\mathcal{V}_1\left([0,T],\mathbb{R}^d\right)\right)$.

The signature of a path is a faithful and universal feature representation and enjoys favourable algebraic and analytic properties, *e.g.*, multiplicative property, characteristic property, and time invariance, etc. These properties distinguish the signature as a useful feature set for time series (*cf.* Lyons et al. (2007); Levin et al. (2013); Chevyrev & Kormilitzin (2016); Kidger et al. (2019); see also Appendix A).

### 2.2 Path Development on matrix Lie groups

Let $G$ be a finite-dimensional Lie group with Lie algebra $\mathfrak{g}$. Assume throughout this paper that $\mathfrak{g}$ is a matrix Lie algebra, namely that a Lie subalgebra of $\mathfrak{gl}(m;\mathbb{F})$, such as the following:

$$\begin{aligned}
\mathfrak{gl}(m;\mathbb{F}) &:= \{m \times m \text{ matrices over } \mathbb{F}\} \cong \mathbb{F}^{m \times m}, \\
\mathfrak{so}(m,\mathbb{R}) = \mathfrak{o}(m,\mathbb{R}) &:= \left\{A \in \mathbb{R}^{m \times m} : A^\top + A = 0\right\}, \\
\mathfrak{sp}(2m,\mathbb{C}) &:= \left\{A \in \mathbb{C}^{2m \times 2m} : A^\top J_m + J_m A = 0\right\}, \\
\mathfrak{su}(m,\mathbb{C}) &:= \left\{A \in \mathbb{C}^{m \times m} : \mathrm{tr}(A) = 0, A^* + A = 0\right\}.
\end{aligned}$$

Here and hereafter, $\mathbb{F} = \mathbb{R}$ or $\mathbb{C}$, and $J_m := \begin{pmatrix} 0 & I_m \\ -I_m & 0 \end{pmatrix}$. In addition, for vector spaces $V_1$, $V_2$, we denote by $\mathbf{L}(V_1, V_2)$ the space of linear transforms $V_1 \to V_2$. For $\mathfrak{gl}(m;\mathbb{F})$ (hence its subsets) we always take the Hilbert–Schmidt norm:

$$\|A\| := \sqrt{\mathrm{tr}(AA^*)} = \sqrt{\sum_{i,j=1}^m \left|A_j^i\right|^2}.$$

**Remark 2.1.** *The Lie group $Sp(2m, \mathbb{C})$ corresponding to the algebra of symplectic matrices $\mathfrak{sp}(2m, \mathbb{C})$ is not compact. In this work we shall sometimes work with*

$$Sp(2m, \mathbb{R}) := Sp(2m, \mathbb{C}) \cap U(2m, \mathbb{C}),$$

*known as the compact symplectic group. It is a* compact real form *of $Sp(2m, \mathbb{C})$, namely that $Sp(2m, \mathbb{R})$ is a compact real Lie group whose Lie algebra $\mathfrak{k}$ satisfies $\mathfrak{k}^{\mathbb{C}} := \mathfrak{k} \otimes_{\mathbb{R}} \mathbb{C} = \mathfrak{sp}(2m, \mathbb{C})$.*

**Definition 2.2** (Path Development). *Fix an integer $m \geq 1$. Let $M : \mathbb{R}^d \to \mathfrak{g} \subset \mathfrak{gl}(m; \mathbb{F})$ be a linear map and let $X \in \mathcal{V}_1\big([0, T], \mathbb{R}^d\big)$ be a path. The path development (*a.k.a. *the Cartan development) of $X$ on $G$ under $M$ is the solution to the equation*

$$\mathrm{d}Z_t = Z_t \cdot M(\mathrm{d}X_t) \qquad \text{for all } t \in [0, T] \text{ with } Z_0 = e, \tag{3}$$

*where $e \in G$ is the group identity and $\cdot$ is the matrix multiplication.*

Write $D_M(X)$ for the endpoint $Z_T$ of the path development of $X$ under $M$.

**Example 2.1.** *For a linear path $X \in \mathcal{V}_1\big([0, T], \mathbb{R}^d\big)$, its development on a matrix Lie group $G$ under $M \in \mathbf{L}(\mathbb{R}^d, \mathfrak{g})$ is*

$$D_M(X)_{0,t} = \exp\big(M(X_t - X_0)\big).$$

*This is because $t \mapsto \exp(M(X_t - X_0))$ is the unique solution to (3).*

**Lemma 2.1** (Multiplicative property of path development). *Let $X \in \mathcal{V}_1\big([0, s], \mathbb{R}^d\big)$ and $Y \in \mathcal{V}_1\big([s, t], \mathbb{R}^d\big)$. Denote by $X * Y$ their concatenation: $(X * Y)(v) = X(v)$ for $v \in [0, s]$ and $Y(v) - Y(s) + X(s)$ for $v \in [s, t]$. Then $D_M(X * Y) = D_M(X)D_M(Y)$ for all $M \in \mathbf{L}\big(\mathbb{R}^d, \mathfrak{g}\big)$.*

By Lemma 2.1 and Example 2.1, the development of piecewise linear paths can be analytically computed. In the example below, the development takes value in the (isometry group of the) hyperboloid. It is a useful tool for studying uniqueness of signature and expected signature (Hambly & Lyons (2010); Boedihardjo et al. (2021); Li & Ni (2022)):

**Definition 2.3** (Hyperbolic Development). *Let $M : \mathbb{R}^2 \to \mathfrak{so}(1, 2)$ be the map $M : (x, y) \mapsto \begin{pmatrix} 0 & 0 & x \\ 0 & 0 & y \\ x & y & 0 \end{pmatrix}$, where $\mathfrak{so}(1, 2)$ is the Lie algebra of the group of orientation-preserving isometries of the hyperbolic plane $\mathbb{H}^2$ in the hyperboloid model:*

$$\mathbb{H}^2 := \big\{x \in \mathbb{R}^3 : (x_1)^2 + (x_2)^2 - (x_3)^2 = -1, \ x_3 > 0\big\}.$$

*Set $p = (0, 0, 1)^T \in \mathbb{H}^2$. The* hyperbolic development *of $X$ is the path*

$$t \longmapsto D_M(X)_{0,t}\, p.$$

The path development shares several properties with signature (summarised below), which are relevant and useful to applications in machine learning. Proofs for Lemma 2.2 and Lemma 2.3 will be provided in Appendix B. The comparison between signature and development will be given at the end of this subsection.

For $M \in \mathbf{L}\big(\mathbb{R}^d, \mathfrak{gl}(m; \mathbb{F})\big)$, we have the canonical extension:

$$\widetilde{M} \in \mathbf{L}\Big(T\big((\mathbb{R}^d)\big), \mathfrak{gl}(m; \mathbb{F})\Big), \quad \widetilde{M}\big(e_{i_1} \otimes e_{i_2} \otimes \cdots \otimes e_{i_k}\big) := M(e_{i_1}) \cdot M(e_{i_2}) \cdot \cdots \cdot M(e_{i_k}). \tag{4}$$

Here $(e_i)_{i=1}^d$ is the standard Cartesian basis for $\mathbb{R}^d$ and $\cdot$ is the matrix multiplication.

**Lemma 2.2** (Link with signature). *Let $X \in \mathcal{V}_1\big([0, T], \mathbb{R}^d\big)$ be a path and $M \in \mathbf{L}\big(\mathbb{R}^d, \mathfrak{g}\big)$ be a linear transform. Then $D_M(X) = \widetilde{M}(S(X))$.*

The development of $X$ can be thought of as $M \mapsto \sum_{k \geq 0} \widetilde{M}\Big(\pi_k\big(S(X)\big)\Big)$, the "generating function" of $S(X)$. Here $\pi_k$ is the projection onto the $k^{\text{th}}$ level of a tensor algebra element.

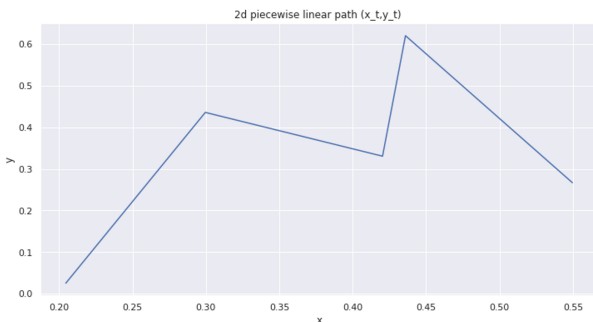 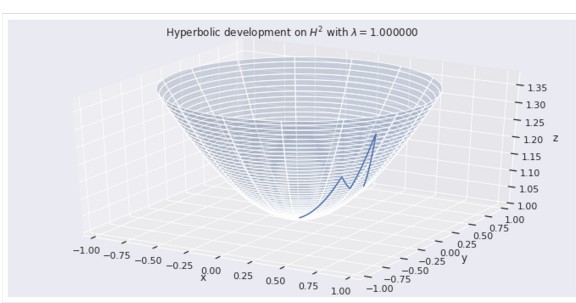

Figure 2: Left panel: A 2-dimensional piecewise linear path $X$; Right panel: the hyperbolic development $D_M(X)$.

**Lemma 2.3** (Invariance under time-reparametrisation)**.** *Let $X \in \mathcal{V}_1\left([0,T], \mathbb{R}^d\right)$ and $\lambda$ be a non-decreasing $\mathcal{C}^1$-diffeomorphism from $[0,T]$ onto $[0,S]$. Define $X_t^\lambda := X_{\lambda_t}$ for $t \in [0,T]$. Then for all $M \in \mathbf{L}\left(\mathbb{R}^d, \mathfrak{g}\right)$ and $s,t \in [0,T]$ we have $D_M(X_{[\lambda_s, \lambda_t]}) = D_M\left(X_{[s,t]}^\lambda\right)$, where $X_{[s,t]}$ denotes $X$ restricted to the interval $[s,t]$.*

**Remark 2.2.** *Lemma 2.3 shows that path development, similar to path signature, is invariant under time re-parameterisation. Thus, the development feature can remove the redundancy caused by the speed of traversing the path, hence bringing about massive dimension reduction and robustness effects to online handwritten character recognition and human action recognition, among other tasks. If, however, the speed information is relevant to the prediction task, one may simply add the time dimension to the input data.*

When choosing adequate Lie algebras, the space of path developments constitutes a rich enough model space to approximate continuous functionals on the signature space.

**Theorem 2.1** (Characteristic property of path development, Theorem 4.8 in Chevyrev & Lyons (2016))**.** *Let $x = (x_0, x_1, \cdots) \in T\left(\left(\mathbb{R}^d\right)\right)$ such that $x_k \neq 0$ for some $k \geq 0$. For any $m \geq \max\{2, k/3\}$, there exists $M \in L\left(\mathbb{R}^d, \mathfrak{sp}(m, \mathbb{C})\right)$ such that $\widetilde{M}(x) \neq 0$. In particular,*

$$D_{\mathrm{sp}}\left(S\left(\mathcal{V}_1\left([0,T], \mathbb{R}^d\right)\right)\right) := \bigcup_{m=1}^\infty \left\{ \widetilde{M} : M \in L\left(\mathbb{R}^d, \mathfrak{sp}(m, \mathbb{C})\right) \right\}$$

*separates points over $S\left(\mathcal{V}_1\left([0,T], \mathbb{R}^d\right)\right)$.*

**Remark 2.3.** *Theorem 2.1 shows that if two signatures differ at $k^{th}$ level, one can find $\widetilde{M} : T\left(\left(\mathbb{R}^d\right)\right) \to G$ that separates the two signatures, whose resulting development has dimension $\leq \left(\max(2, \frac{k}{3})\right)^2$. This number is typically much smaller than the dimension of the truncated signature up to degree $k$, namely $\frac{d^{k+1}-1}{d-1}$.*

Chevyrev & Lyons (2016) also proved that the unitary representation of path development is universal to approximate continuous functionals on the signature space. One may refer to Theorem B.1 in the Appendix.

## 2.3 Optimisation on Lie Group

Now we briefly discuss a gradient-based optimisation method on Riemannian manifolds introduced in Lezcano-Casado (2019); Lezcano-Casado & Martınez-Rubio (2019). We shall focus on matrix Lie groups; see Lezcano-Casado (2019) for general manifolds.

**Definition 2.4** (Trivialisation)**.** *Let $G$ be a matrix Lie group. A surjection $\phi : \mathbb{R}^d \to G$ is said to be a trivialisation.*

Trivialisations allow us to reduce an optimisation problem constrained on a manifold to an unconstrained problem on a vector space, namely that

$$\min_{x \in G} f(x) \rightsquigarrow \min_{a \in \mathbb{R}^d} f(\phi(a)).$$

The right-hand side can be solved numerically by gradient descent once $\nabla(f \circ \phi)$ is computed.

For a compact connected matrix group $G$ with Lie algebra $\mathfrak{g}$, the matrix exponential exp serves as the "canonical" trivialisation. For the noncompact case where exp is not surjective, it can still be used as a "local trivialisation". See Theorem 4.7 in Lezcano-Casado (2019).

The formulae in Theorem 2.2 below enable fast numerical implementation of the gradient $\nabla(f \circ \exp)$ over matrix groups.

**Theorem 2.2.** *Let $f : \mathfrak{gl}(m; \mathbb{C}) = \mathbb{C}^{m \times m} \to \mathbb{R}$ be a scalar function, let $\exp$ be the matrix exponential, and let $A$ be any matrix in $\mathfrak{gl}(m; \mathbb{C})$. We have the following*

- *formula for gradient (Lezcano-Casado (2019)):*

$$\nabla(f \circ \exp)(A) = (\mathrm{d}\exp)_{A^\top}\left(\nabla f\big(\exp(A)\big)\right);$$

- *formula for the differential of exponential (Rossmann (2002)):*

$$(\mathrm{d}\exp)_A(X) = \sum_{k=0}^{\infty} \frac{\big(-\mathrm{ad}(A)\big)^k}{(k+1)!}\big(\exp(A)X\big),$$

*where $\mathrm{ad}$ is the adjoint action $\mathrm{ad}(X)Y := XY - YX$.*

A variant of the first formula in Theorem 2.2 is particularly useful for our numerical experiments in §4. Here and hereafter, denote the *nearest point projection* from $\mathfrak{gl}(m, \mathbb{C})$ to a given Lie subalgebra $\mathfrak{g}$ as

$$\mathbf{Proj} : \mathfrak{gl}(m, \mathbb{C}) \longrightarrow \mathfrak{g}. \tag{5}$$

For instance, $\mathbf{Proj}(M) = \frac{1}{2}\left(M - M^\top\right)$ for $\mathfrak{g} = \mathfrak{so}(m, \mathbb{R})$. We then have

**Lemma 2.4** (Gradient on matrix Lie algebra)**.** *Let $f : \mathfrak{gl}(m, \mathbb{C}) \to \mathbb{R}$ be a smooth scalar field and $\mathbf{Proj} : \mathfrak{gl}(m, \mathbb{C}) \to \mathfrak{g}$ be the projection as before. Then*

$$\nabla(f \circ \exp \circ \mathbf{Proj})(A) = \mathbf{Proj}\Big\{\mathrm{d}_{M^\top}\exp\Big(\nabla\big(f \circ \exp(M)\big)\Big)\Big\},$$

*where $A \in \mathfrak{gl}(m; \mathbb{C})$ and $M = \mathbf{Proj}(A)$. Here we use $\exp$ to denote both the exponential maps $\mathfrak{g} \to G$ and $\mathfrak{gl}(m, \mathbb{C}) \to \mathrm{GL}(m, \mathbb{C})$.*

*Proof.* For each $A \in \mathfrak{gl}(m, \mathbb{C})$ it holds that

$$\nabla_A(f \circ \exp \circ \mathbf{Proj}) = \mathbf{Proj}\big(\nabla_M(f \circ \exp)\big),$$

where $M = \mathbf{Proj}(A) \in \mathfrak{g} \subset \mathfrak{gl}(m, \mathbb{C})$. Moreover,

$$\nabla_M(f \circ \exp) = \mathrm{d}_{M^\top}\exp\Big(\nabla\big(f \circ \exp(M)\big)\Big)$$

due to Proposition 6.1 in Lezcano-Casado (2019). $\qquad\square$

## 3   Path Development Layer

The main objective of this paper is to propose a novel neural network layer, termed as the *path development layer*. It is a generic, trainable module for time series data.

### 3.1 Network architecture

In practice, one often observes discrete time series. Let $x = (x_0, \cdots, x_N) \in \mathbb{R}^{d \times (N+1)}$ be a $d$-dimensional time series. Once the Lie algebra $\mathfrak{g}$ and $M \in \mathbf{L}(\mathbb{R}^d, \mathfrak{g})$ are specified, one can define the development of $x$ under $M$ via the continuous linear interpolation of $x$. We parameterise the linear map $M \in \mathbf{L}(\mathbb{R}^d, \mathfrak{g})$ of the development $\mathcal{D}_M$ by its linear coefficients $\theta \in \mathfrak{g}^d$. That is, given $\theta = (\theta_1, \cdots, \theta_d) \in \mathfrak{g}^d$, define

$$M_\theta : \mathbb{R}^d \ni x = \left(x^{(1)}, \cdots, x^{(d)}\right) \longmapsto \sum_{j=1}^d x^{(j)} \theta_j \in \mathfrak{g}.$$

The path development layer will be constructed below in a recursive fashion.

**Definition 3.1** (Path development layer)**.** *Fix a matrix group $G$ with Lie algebra $\mathfrak{g}$. The path development layer is defined as a map $\mathcal{D}_\theta : \mathbb{R}^{d \times (N+1)} \to G^{N+1}[or\ G, resp.] : x = (x_0, \cdots, x_N) \mapsto z = (z_0, \cdots, z_N)$ [or $z_N$, resp.] such that for each $n \in \{0, \cdots, N-1\}$,*

$$z_{n+1} = z_n \exp\left(M_\theta(x_{n+1} - x_n)\right), \quad z_0 = \mathbf{Id}_m. \tag{6}$$

*Here $\exp$ is the matrix exponential, $G^{N+1}$ is the $(N+1)$-fold Cartesian product of $G$, and $\theta \in \mathfrak{g}^d$ constitutes trainable model weights.*

The path development layer is designed to take the form of Eq. (6) mainly for two reasons: (1), the multiplicative property of development (Lemma 2.1); (2) the explicit solution for linear paths (Example 2.1). Notice too that the output of development at step $n$ admits an analytic formula:

$$z_n = \exp(M_\theta(x_1 - x_0)) \exp(M_\theta(x_2 - x_1)) \cdots \exp(M_\theta(x_n - x_{n-1})) \in G.$$

**Remark 3.1** (Connection with RNN)**.** *The recurrence structure of the development in Eq. (6) resembles that of the recurrent neural network (RNN); the development output $z_n$ plays the role as the hidden neuron of RNNs $h_n$. In contrast, our path development layer does not require a fully connected neural network to construct the hidden neurons.*

*ExpRNN (Lezcano-Casado (2019)), a geometric variant of RNN model, also possesses the built-in Lie group structure and restricts the model parameters $\theta$ to the orthogonal Lie group in the recurrence relation $h_{n+1} = \sigma(\theta h_n + T x_{n+1})$. The hidden neurons of ExpRNNs, as opposed to the path development layer, may fail to live in the Lie group.*

**Remark 3.2** (Comparison with path signature layer)**.** *The signature layer of the path maps time series $x = (x_0, \cdots, x_N)$ to $s = (s_0, \cdots, s_N)$ according to the following equation, which is of a similar form to Eq. (6):*

$$s_n = s_{n-1} \otimes \exp(x_{n+1} - x_n) \text{ for all } n \in \{1, \cdots, N\}; \qquad s_0 = \mathbf{1}.$$

*In comparison with the deterministic signature layer, the path development has trainable weights. When choosing the matrix Lie algebra $\mathfrak{g} \subset \mathfrak{gl}(m, \mathbb{F})$, the resulting development feature lies in $GL(m, \mathbb{F})$. Note that $\dim_{\mathbb{F}} GL(m, \mathbb{F}) = m^2$, independently of the path dimension $d$. This is in stark contrast with the signature, whose range (at truncated order $n$) has dimension depending geometrically on $d$.*

The development layer allows for both sequential output $z = (z_i)_{i=0}^N \in G^{N+1}$ and static output $z_N \in G$. Its forward evaluation is summarised in Algorithm 1 below.

---

**Algorithm 1** Forward Pass of Path Development Layer

---

1: **Input:** $\theta \in \mathfrak{g}^d \subset \mathbb{R}^{m \times m \times d}$ (model parameters), $x = (x_0, \cdots, x_N) \in \mathbb{R}^{d \times (N+1)}$ (input time series), $m \in \mathbb{N}$ (order of the matrix Lie algebra), $(d, N)$ are the feature and time dimensions of $x$, respectively.
2: $z_0 \leftarrow \mathrm{Id}_m$
3: **for** $n \in \{1, \cdots N\}$ **do**
4: $\quad z_n \leftarrow z_{n-1} \exp(M_\theta(x_n - x_{n-1}))$
5: **end for**
6: **Output:** $z = (z_0, \cdots, z_N) \in G^{N+1} \subset \mathbb{R}^{m \times m \times (N+1)}$ (sequential output) or $z_N \in G \subset \mathbb{R}^{m \times m}$ (static output).

---

### 3.2 Model parameter optimisation

We shall utilise a method introduced in Lezcano-Casado (2019); Lezcano-Casado & Martınez-Rubio (2019) to optimise model parameters of the development layer, which effectively leverages the Lie group-valued outputs and facilitates efficient gradient computation by exploring the recurrence structure via backpropagation through time.

More specifically, consider a scalar field $\psi : G^{N+1} \to \mathbb{R}$ and an input $x := (x_n)_{n=0}^N \in (\mathbb{R}^d)^{N+1}$. The goal is to seek optimal parameters $\theta^*$ minimising $\psi(\mathcal{D}_\theta(x))$; in formula,

$$\theta^* = \operatorname{argmin}_{\theta \in \mathfrak{g}^d} \psi(\mathcal{D}_\theta(x)),$$

where $\mathcal{D}_\theta$ is the development layer in Eq. (6).

Due to the Lie group structure of the output, the gradient descent in Euclidean spaces is not directly applicable here. We adapt the method of trivialisation (Lezcano-Casado (2019); Lezcano-Casado & Martınez-Rubio (2019)) to do gradient computations. Taking into account the recurrence structure of the development, we express the gradients of the development layer in a form similar to the Recurrent Neural network, and thus implement the corresponding backpropagation through time algorithm for parameter optimisation.

To describe the optimisation method on Lie groups in full detail, first let us fix some notations. Recall that $z := (z_n)_{n=0}^N \in G^{N+1}$ denotes the output of the development layer $\mathcal{D}_\theta$, whose input is $x = (x_n)_{n=0}^N \in \mathbb{R}^{d \times (N+1)}$. The variables $z_n$ have the recursive structure $z_n = z_{n-1} \cdot \exp(M_\theta(\Delta x_n))$; see Eq. (6). We introduce the *Step-i update function*:

$$\mathfrak{S}_i : G \times \mathfrak{g}^d \longrightarrow G, \qquad (z, \theta) \longmapsto z \exp(M_\theta(\Delta x_i)).$$

The output of development layer $(z_i)_{i=0}^N$ can be expressed by the update function:

$$z_i = \mathfrak{S}_i(z_{i-1}, \theta) \text{ for each } i \in \{1, \cdots, N\}; \qquad z_0 = \mathbf{Id}_m.$$

When there is no ambiguity, we simply write $\mathfrak{S}_i(z_{i-1})$ for $\mathfrak{S}_i(z_{i-1}, \theta)$.

Denote by $\mathbf{pr}_i$ the projection of $G^{N+1}$ onto the $i^{\text{th}}$ coordinate. This notation is selected to distinguish it from $\pi_k$, the $k^{\text{th}}$-level truncation of the signature (Eq. (8)), or the projection $\mathbf{Proj}$ from square matrices to $\mathfrak{g}$ (Eq. (5)). For a scalar field $\psi : G^{N+1} \to \mathbb{R}$, the correct definition for its "$i^{\text{th}}$ partial derivative" is $(\mathbf{pr}_i)_\# \, \mathrm{d}\psi$, the pushforward of $\mathrm{d}\psi$ via $\mathbf{pr}_i$:

$$(\mathbf{pr}_i)_\# \, \mathrm{d}\psi \Big|_z : T_{\mathbf{pr}_i(z)} G \to T_{\psi(z)} \mathbb{R} \cong \mathbb{R} \qquad \text{for each } z \in G^{N+1}.$$

For each $n \in \{0, 1, \ldots, N-1\}$ we also introduce the function $\widetilde{\psi}_n : G \to \mathbb{R}$ as follows:

$$\widetilde{\psi}_n(z_n) := \psi\Big(z_0, \, \ldots, \, z_n, \, \mathfrak{S}_{n+1}(z_n), \, \mathfrak{S}_{n+2} \circ \mathfrak{S}_{n+1}(z_n), \, \ldots, \, \mathfrak{S}_N \circ \cdots \circ \mathfrak{S}_{n+1}(z_n)\Big). \tag{7}$$

Recall from (6) that $z_n$, the output of development layer, is defined recursively with respect to $n$. As a consequence, the exterior differential of $\widetilde{\psi}_n$ also has a recursive structure.

**Theorem 3.1** (Recursive structure of the differential at $z_n$). *Let $x = (x_0, \cdots, x_N) \in (\mathbb{R}^d)^{N+1}$ and $z = (z_0, \cdots, z_N) \in G^{N+1}$ be the input and output of the development layer as before, where $G$ is a matrix Lie group. Let $\psi : G^{N+1} \to \mathbb{R}$ be a smooth scalar field and $\widetilde{\psi}_n$ be as in Eq. (7). For each $n \in \{1, \cdots, N\}$ we can express the 1-form $\mathrm{d}\widetilde{\psi}_n$ on $G$ as follows:*

$$\mathrm{d}_{z_n} \widetilde{\psi}_n(\xi \cdot z_n) = (\mathbf{pr}_n)_\# \, \mathrm{d}\psi \Big|_z (z_n \cdot \xi) + \mathrm{d}_{z_{n+1}} \widetilde{\psi}_{n+1}\Big(\xi \cdot z_{n+1}\Big) \quad \text{for each } \xi \in \mathfrak{g}.$$

*Here $\Delta x_{n+1} := x_{n+1} - x_n$ and $\cdot$ is the matrix multiplication.*

In the above $\mathrm{d}\widetilde{\psi}_n : T_{z_n} G \to T_{\widetilde{\psi}_n(z_n)} \mathbb{R} \cong \mathbb{R}$. A generic element of the domain $T_{z_n} G$ takes the form $z_n \cdot \xi \equiv \mathrm{d}\mathcal{L}_{z_n}(\xi)$ for $\xi \in \mathfrak{g} = T_{\mathrm{Id}} G$, where $\mathcal{L}_{z_n} : G \to G$ is the left multiplication. One may view $\mathrm{d}\widetilde{\psi}_n$ as a 1-form on $G$ or, equivalently, as an element of $\mathfrak{g}^*$.

By virtue of Theorem 3.1 and the duality between gradient and differential, we are able to determine the gradient of $\widetilde{\psi}_n$, which is the main ingredient of the Riemannian gradient descent algorithm of the development layer (*i.e.*, Algorithm 2 below). Before further developments, let us first comment on the Riemannian gradient on matrix Lie groups.

**Remark 3.3.** *Given a scalar field* $f : (\mathcal{M}, g) \to \mathbb{R}$, *recall that its gradient* $\nabla f$ *is the vector field on* $\mathcal{M}$ *determined by*

$$g(\nabla f, V) = \mathrm{d}f(V)$$

*for any vector field* $V$.

Throughout this paper, when $\mathcal{M} = \mathfrak{gl}(m, \mathbb{F})$ for $\mathbb{F} = \mathbb{R}$ or $\mathbb{C}$, we always take the Riemannian metric given by the Hilbert–Schmidt norm of matrices.

**Proposition 3.1** (Gradient with respect to model parameter $\theta$). *Let* $x \in \left(\mathbb{R}^d\right)^{N+1}$, $z \in G^{N+1}$, *and* $\theta \in \mathfrak{g}^d$ *be the input, output, and model parameter of the development layer* $\mathcal{D}_\theta : \left(\mathbb{R}^d\right)^{N+1} \to G^{N+1}$, *respectively. The gradient of* $\theta \mapsto \psi \circ \mathcal{D}_\theta(x)$ *for any scalar field* $\psi : G^{N+1} \to \mathbb{R}$ *is determined by the expression*

$$\nabla\big(\psi \circ \mathcal{D}_\theta(x)\big) = \sum_{n=1}^N \nabla_\theta\Big(\widetilde{\psi}_n \circ \mathfrak{S}_n(z_{n-1}, \theta)\Big),$$

*where* $\mathfrak{S}_n$ *is the Step-n update function and* $\widetilde{\psi}_n$ *is the updated version of as in Eq. (7). Furthermore, the right-hand side can be computed via*

$$\nabla_\theta\Big(\widetilde{\psi}_n \circ \mathfrak{S}_n(z_{n-1}, \theta)\Big) = \mathbf{Proj}\Big(\mathrm{d}_{[M_\theta(\Delta x_n)]^\top} \exp\Big\{\big[\nabla_{z_n}\widetilde{\psi}_n\big]\big(M_\theta(\Delta x_n)\big) \cdot z_{n-1}\Big\}\Big).$$

With the gradient computation at hand (in particular, Theorem 3.1 and Proposition 3.1), we are now ready to describe the backpropagation of development layer in Algorithm 2.

---

**Algorithm 2** Backward Pass of Path Development Layer

---

1: **Input:** $x = (x_0, \cdots, x_N) \in \mathbb{R}^{(N+1)\times d}$ (input time series), $z = (z_0, \cdots, z_N) \in G^{N+1}$ (output series by the forward pass), $\theta = (\theta_1, \cdots, \theta_d) \in \mathfrak{g}^d \subset \mathbb{R}^{m \times m \times d}$ (model parameters), $\eta \in \mathbb{R}$ (learning rate), $\psi : G^{N+1} \to \mathbb{R}$ (loss function).
2: Initialize $a \leftarrow 0$             ▷ $a$ represents $\mathrm{d}_{z_n}\widetilde{\psi}_n$
3: Initialize $\omega \leftarrow 0$          ▷ $\omega$ represents $\nabla_\theta\left(\psi \circ \mathcal{D}_\theta(x)\right)$
4: **for** $n \in \{N, \cdots, 1\}$ **do**
5:    Compute $\mathfrak{m} \leftarrow \exp\big(M_\theta(\Delta x_n)\big)$
6:    Compute $a \leftarrow (\mathbf{pr_n})_\# \mathrm{d}\psi\big|_{y=z} + a \cdot \mathfrak{m}$       ▷ by Theorem 3.1.
7:    Compute $\omega \leftarrow \omega + \mathrm{d}_{\mathfrak{m}^\top} \exp(a) \otimes \Delta x_n$    ▷ by Proposition 3.1 and Theorem 2.2.
8: **end for**
9: $\theta \leftarrow \theta - \eta \cdot \omega$.
10: **for** $i \in \{1, \cdots, d\}$ **do**
   $\theta_i \leftarrow \mathbf{Proj}\left(\theta_i\right) \in \mathfrak{g}$.            ▷ by Equation (5).
11: **end for**
12: **Output:** return $\theta$ (updated model parameter).

---

One crucial remark is in order here. In view of Theorem 3.1 and Proposition 3.1, the development layer proposed in this paper possesses a recurrence structure analogous to that of the RNNs. This is the key structural feature of Algorithm 2. However, it is well known that the RNNs are, in general, prone to problems of vanishing and/or exploding gradients (see Bengio et al. (1994)). We emphasise that *when $G$ is the orthogonal or unitary group*, the gradient issues are naturally alleviated for the development layer.

**Remark 3.4.** *The differential of the update function is an isometry when* $G = SO(m)$ *or* $U(m)$. *That is, when the matrix group $G$ is equipped with the Hilbert–Schmidt norm,* $\mathrm{d}\mathfrak{S}_i : TG \to TG$ *has operator norm*

1 *for any $i \in \{0, 1, \ldots, N-1\}$ and $z \in G^{N+1}$. (Here $\mathrm{d}\mathfrak{S}_i$ is understood as $\mathrm{d}\mathfrak{S}_i(\bullet, \theta_i)$ for fixed $\theta_i \in \mathfrak{g}^d$; we suppress the dependence on $\theta_i$ for simplicity.) This is because*

$$\mathrm{d}_{z_i}\mathfrak{S}_i\big((\mathcal{L}_{z_i})_\# \eta\big) = z_i \cdot \eta \cdot \exp\big(M_{\theta_{i+1}}\Delta x_{i+1}\big) \qquad \text{for any } \eta \in \mathfrak{g}$$

*and, on the right-hand side, both the left multiplication by $z_i$ and the right multiplication by $\exp\big(M_{\theta_{i+1}}\Delta x_{i+1}\big) \in G$ are isometries for $G = O(m)$ or $U(m)$. More concretely,*

$$\big\|\mathrm{d}_{z_i}\mathfrak{S}_i\big((\mathcal{L}_{z_i})_\#\eta\big)\big\| = \|\eta\| \qquad \text{for any } \eta \in \mathfrak{g}.$$

The implementation of the development layer can be flexibly adapted to general matrix Lie algebras via **Proj** in Eq. (5). The corresponding Lie groups used in our implementation include the special orthogonal, unitary, real symplectic, and special Euclidean groups, as well as the group of orientation-preserving isometries of the hyperbolic space.

We adopt `https://github.com/Lezcano/expRNN` for the PyTorch implementation of matrix exponential and its differential as in Al-Mohy & Higham (2009; 2010) to efficiently evaluate and train the development layer. A scaling-and-square trick for matrix exponential computation (Higham (2005)) is used to facilitate stabilisation of model optimisation. To sum up, the computation of Algorithms 1 and 2 requires $\mathcal{O}(N)$ time and $\mathcal{O}(N)$ storage complexity ($N$ = the length of time series). The complexity of Algorithm 1 may be optimised by pre-computing the Lie group-valued path increments in parallel and then computing the path development output iteratively.

## 4 Numerical experiments

We begin this section with validating the model performance of the proposed development layer (DEV) and the hybrid model, constructed by stacking LSTM with the development layer (LSTM+DEV), for general sequential data tasks in Section 4.1. We then proceed with two examples of simulated data to articulate its capability of modelling trajectories on non-Euclidean spaces in Section 4.2, which may fail to be accomplished by RNN or its existing geometric variants (*e.g.*, ExpRNN).

In the subsequent experiments, we consider the development with special orthogonal, real symplectic, and special Euclidean groups, denoted as DEV(SO), DEV(Sp), and DEV(SE) respectively. The standard loss functions are chosen: cross entropy for the time series classification tasks in Section 4.1 and mean squared error (MSE) for the regression tasks in Section 4.2. Full implementation details of the experiments can be found in Appendix D. Additionally, we have included the relevant codes in the supplementary material to ensure reproducibility.

### 4.1 Time series and sequential image modelling

The path development network with suitably chosen matrix Lie group has a broad range of capabilities (*e.g.*, capturing long-term dependencies, handling irregular time series, improving training stability and convergence rate) for a wide range of applications (e.g. character trajectories, audio, images). This is demonstrated using the following datasets. (1) Character Trajectories (Bagnall et al. (2018)). (2) Speech Commands dataset (Warden (2018)). (3) Sequential MNIST(Le et al. (2015)), permuted sequential MNIST (Le et al. (2015)), and sequential CIFAR-10 (Chang et al. (2017)).

We benchmark the proposed development models with various models, including RNN-based and/or continuous time series models (*e.g.*, LSTM, NCDE, GRU-ODE) as baselines and state-of-the-art (SOTA) models for each task.

### 4.1.1 Speech Command dataset

We first demonstrate the performance of the path development network on the Speech Commands dataset (Warden (2018)) as an example of high-dimensional long time series. We follow Kidger et al. (2020) to precompute mel-frequency cepstrum coefficients of the audio recording and consider a balanced classification

task of predicting a spoken word. This processed dataset consists of 34975 samples of secondly audio recordings, represented in a regularly spaced time series of length 169 and path dimension 20.

Table 1: Test accuracies (%) of the linear models using multiple orthogonal path developments on the Speech Commands dataset *w.r.t.* the different matrix sizes $m$ and number of path development layers $N$.

| Matrix size | Number of developments | | | |
|---|---|---|---|---|
| | 1 | 2 | 4 | 8 |
| 5 | 70.0 | 77.3 | 81.5 | 83.4 |
| 10 | 81.4 | 83.8 | 84.3 | 85.2 |
| 20 | 84.4 | 86.0 | **86.5** | 85.6 |

**Path development as a universal representation with dimension reduction.** We apply linear models on both path signature and path development to Speech Command and Character trajectories datasets. Both features achieve high performance on their own, thus validating universality (see Theorem B.1). As shown in Table 1, increasing the number of developments or the matrix size leads to an improvement in classification accuracy in general. The only exception is that for matrix size 20, the accuracy is reduced by 0.9% if we increase the number of developments from 4 to 8, which may result from overfitting. In Figure 3, the curve of test accuracy against feature dimension of the single development (Red) is consistently above that of the signature (Blue), demonstrating that development is a more compact feature than signature. The development layer with the matrix group of order 30 achieves an accuracy about 86% of that of the signature, but with dimension reduced by a factor of 9 (900 *v.s.* 8420).

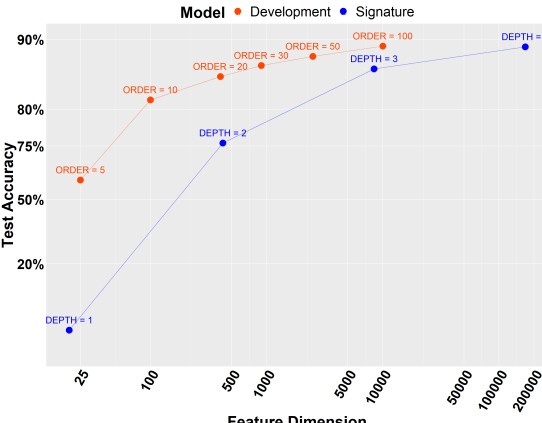

Figure 3: Test accuracy v.s. the feature dimension curves of the linear models using the development and signature representation on Speech Commands dataset. See full results in Table 6.

**State-of-the-art performance.** We first compare the proposed development model with other continuous time series model baselines, including GRU-ODE (De Brouwer et al. (2019)), ODE-RNN (Rubanova et al. (2019)), and NCDE (Kidger et al. (2020)). Our model significantly outperforms GRU-ODE and ODE-RNN on this task, whilst achieve comparable test accuracy with NCDE. The average accuracy of NCDE is 1.9% higher than our development model. However, the standard deviation of the NCDE model is 2.5%, higher than that of the development model (0.1%), thus making the accuracy difference of these two models insignificant. Note that NCDEs fully parameterize the vector field of the controlled differential equations (CDEs) by a neural network, while our proposed development satisfies *linear* CDEs with trainable weights constrained in the matrix Lie algebra. The difference in complexities of the relevant CDEs may contribute to the higher average test accuracy and higher variation of the prediction by NCDE.

Compared with the strong baseline LSTM, development layer alone has a lower accuracy by 7.8%. However, our hybrid model (LSTM+DEV) further improves the LSTM performance by 0.9%. The LSTM+DEV model achieves 96.8% test accuracy, which is slightly lower than that of the state-of-the-art FLEXTCN (Romero et al. (2021a)). But, involving only a quarter of parameters of the latter, our model is significantly more compact.

Table 2: Test accuracies (%) (mean ± std, computed across 5 runs) on Speech Commands dataset

| Model | Test accuracy(%) | # Params(K) |
|---|---|---|
| ODE-RNN Rubanova et al. (2019) | 65.9 ±35.6 | 89 |
| NCDEKidger et al. (2020) | 89.8 ±2.5 | 89 |
| LSTM | 95.7 ±0.2 | 88 |
| EXPRNN Lezcano-Casado & Martınez-Rubio (2019) | 82.1 | 270 |
| LipschitzRNN Erichson et al. (2020) | 88.38 | 270 |
| CKConv Romero et al. (2021b) | 95.3 | 100 |
| FlexTCNRomero et al. (2021a) | **97.7** | 373 |
| S4 Gu et al. (2021a) | 95.3 % | 260 |
| LSSLGu et al. (2021b) | 93.58 | 330 |
| Signature | 85.7±0.1 | 84 |
| DEV(SO) | 87.9±0.1 | 87 |
| LSTM+DEV(SO) | 96.8±0.1 | 86 |

### 4.1.2 Character Trajectories data

Next we investigate the performance of the development network on *irregular* time series in terms of accuracy, robustness and training stability. We consider the Character Trajectories from the UEA time series classification archive (Bagnall et al. (2018)). This dataset consists of Latin alphabet characters written in a single stroke with their 2-dimensional positions and pen tip force. There are 2858 time series with a constant length of 182 and 20 unique characters to classify. Following the approach in Kidger et al. (2020), we randomly drop 30%, 50%, or 70% of the data in the same way for every model and every repeat, which results in irregular sampled time series. In addition, we validate the robustness of model to timescale change following Gu et al. (2020).

**State-of-the-art accuracy.** Similar to Speech Commands dataset, the development layer itself outperforms the signature layer, although still underperforms NCDE and LSTM in terms of test accuracy. However, the proposed hybrid model (LSTM+DEV) with special orthogonal group achieves the state-of-the-art accuracy, with around 0.6-0.7% and 0-1.1% performance gain in comparison to the best NCDE and CKCNN models, respectively.

**Robustness.** The path development layer exhibits robustness and improves the robustness of LSTM in the following two settings: (1) data dropping; (2) timescale distribution shift.
*Robustness to dropping data.* Table 3 demonstrates that, as observed for other continuous time series models, the accuracy of DEV and LSTM+DEV remains similar across different drop rates, thus exhibiting robustness against data dropping and irregular sampling of time series. LSTM achieves competitive accuracy of ∼94% when the drop rate is relatively low, with a drastic performance decrease (6%) for the high drop rate (70%). However, by adding the development layer, especially the one with the special orthogonal group, one significantly improves the test accuracy of LSTM by 11.4% to a high drop rate (70%).
*Robustness to timescale distribution shift.* We follow the approach in Gu et al. (2020), where the train and test character trajectories are sampled at different timescales. Table 3 shows that, while all the baselines suffer from severe deterioration in test accuracy in this setting, the path development layer retains high performance, similar to HIPPO (Gu et al. (2020)) and improves the accuracy of LSTM by ∼60%. This is because the path development is invariant under time-reparameterisation (Lemma 2.3).

Table 3: Test accuracies(%) on the Character Trajectories dataset for different data dropping rate and time scale distribution shifts. ∗ means taken from our reproduced baseline models.

| Model | Test accuracy(%) | | | | | |
|---|---|---|---|---|---|---|
| Task | Drop rate | | | Sampling rate | | |
| | 30% | 50% | 70% | $1 \to 1$ | $1/2 \to 1$ | $1 \to 1/2$ |
| GRU-ODE De Brouwer et al. (2019) | 92.6 ±1.6 | 87.6 ±3.9 | 89.9 ±3.7 | 96.2 | 23.1 | 25.5 |
| NCDE Kidger et al. (2020) | 98.7 ±0.8 | 98.8 ±0.2 | 98.6 ±0.4 | 98.8 | 44.7 | 11.3 |
| LSTM | 94.0 ± 3.9 | 94.6 ± 2.0 | 87.8 ± 6.3 | 91.3 | 31.9 | 28.2 |
| ExpRNN Lezcano-Casado & Martinez-Rubio (2019) | 95.8 ± 0.7 | 96.6 ± 0.9 | 95.6±0.7 | 97.2∗ | 17.3∗ | 11.0∗ |
| CKConv Romero et al. (2021b) | **99.3** | 98.8 | 98.1 | - | - | - |
| HIPPO Gu et al. (2020) | - | - | - | - | 88.8 | 90.1 |
| Signature | 92.1 ±0.2 | 92.3 ± 0.5 | 90.2±0.4 | 92.9 | 92.3 | 91.5 |
| DEV (SO) | 92.7± 0.6 | 92.6±0.9 | 91.9 ±0.9 | 93.2 | 91.6 | **92.3** |
| LSTM+DEV (SO) | **99.3± 0.3** | **99.3 ± 0.1** | **99.2± 0.3** | **99.5** | **94.6** | 91.3 |
| LSTM+DEV (Sp) | 97.7± 0.2 | 97.7 ± 0.2 | 97.4± 0.4 | 97.9 | 94.2 | 89.3 |

Figure 4: The comparison plot of LSTM and LSTM+DEV on Character Trajectories with 30% drop rate. (Left) The evolution of validation loss against training time. (Middle) The evolution of validation accuracy against training time. The mean curve with ± std indicated by the shaded area is computed over 5 runs. (Right) The boxplot of the validation accuracy for varying learning rate.

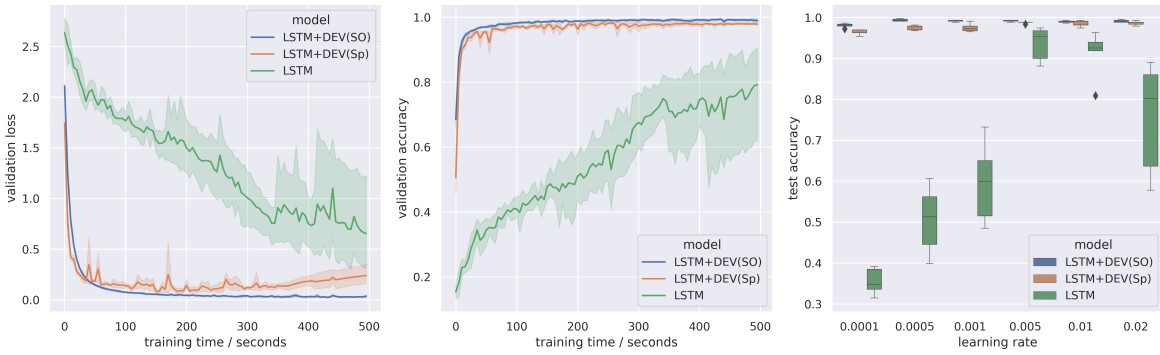

**Improvement of training LSTMs by Development Layer (LSTM+DEV).** Apart from model robustness and compactness, the additional development layer improves the LSTM in terms of (1), training stability and convergence; and (2), less need for hyper-parameter tuning. This has led to better model performances with faster training.

***Training stability and convergence.*** The left and middle subplots of Figure 4 show the oscillatory evolution of the loss and accuracy of the LSTM model during training, which can be significantly stabilised by adding DEV(SO) and DEV(Sp). In particular, the performance of DEV(SO) is better than that of DEV(Sp) in terms of both training stability and prediction accuracy. It suggests that the development layer may facilitate the saturation of hidden states of LSTMs and resolving the gradient vanishing/exploding issues, by virtue of the boundedness of gradients resulting from $|\partial_{z_{n-1}} z_n| = 1$ of DEV(SO) (Remark 3.4). It can thus serve as an effective means for alleviating long temporal dependency issues of LSTM. Moreover, our hybrid model has a much faster training convergence rate than LSTM.

***Sensitivity of learning rate.*** Figure 4 (Right) shows that the LSTM performance is highly sensitive to the learning rate. So, extensive, time-consuming hyper-parameter tuning is a must to ensure high performance of LSTM. However, the LSTM+DEV model training is much more robust to the learning rate, leading to less hyper-parameter tuning and consistently superior performance over LSTM.

### 4.1.3 Sequential image classification

We consider three time series datasets of large scale for the sequential image classification: sequential MNIST(Le et al. (2015)), permuted sequential MNIST (Le et al. (2015)), and sequential CIFAR-10 (Chang et al. (2017)), consisting of sequences of pixel-by-pixel with lengths 784 and 1024, respectively. Each has 50000 training samples and 10000 test samples.

**Accuracy comparison and model compactness.** On the sequential MNIST and CIFAR-10 datasets, stacking a path development layer of a $10 \times 10$ matrix to the LSTM significantly increases the performance of the baseline LSTM by 7–11% . Among RNN-based models, the hybrid model achieves similar results as the state-of-the-art Lipschitz RNN model, but with only 1/5 model parameters.

In comparison with the state-of-the-art models FlexTCN (Romero et al. (2021a)), S4 (Gu et al. (2021a)), and LSSL (Gu et al. (2021b)), the proposed LSTM+DEV model slightly underperforms on the sequential MINST and permuted MNIST data by 0.7% and 2.6% respectively. However, note it only uses 72k model parameters compared with a larger number of model parameters required for sota models, i.e., 375k (FlexTCN), 6000k (S4) and 200k (LSSL). However, there is still a significant gap between LSTM+DEV and the stoa models on the Cifar10 dataset, which merits further investigation in future.

Table 4: Test accuracies (%) of the sequential image classification, ∗ indicates the results from our reproduced baseline models

| Model | Test accuracy (%) | | | Time (s/epoch) | Memory(Mb) | Params(k) |
|---|---|---|---|---|---|---|
| Dataset | sMNIST | pMNIST | sCIFAR-10 | sMNIST/sCIFAR-10 | | |
| LSTMBai et al. (2018) | 87.2 | 85.7 | 57.6∗ | 25/29∗ | 1638/1798∗ | 70 |
| r-LSTM Trinh et al. (2018) | 98.4 | 95.2 | 72.2 | - | - | 500 |
| DTRIVLezcano-Casado (2019) | 98.9 | 96.5 | 30.0∗ | 120/131∗ | 1887/2183∗ | 130 |
| EXPRNNLezcano-Casado & Martinez-Rubio (2019) | 98.4 | 96.2 | 35.9∗ | 119/125∗ | 1887/2183∗ | 130 |
| LipschitzRNN Erichson et al. (2020) | 99.4 | 96.3 | 64.2 | - | - | 260 |
| CKConv Romero et al. (2021b) | 99.3 | 98.0 | 62.3 | - | - | 98 |
| FlexTCN Romero et al. (2021a) | **99.6** | 98.6 | 80.8 | - | - | 375 |
| S4 Gu et al. (2021a) | **99.6** | 98.7 | **91.1** | - | - | 6000 |
| LSSLGu et al. (2021b) | 99.5 | **98.8** | 84.7 | - | - | 200/2000 |
| LSTM+DEV(SO) | 98.9± 0.2 | 96.2 ± 0.3 | 64.3± 0.9 | 62/63 | 1957/2207 | 72 |

**Training time & Memory.** The training time of the development layer is comparable to that of LSTM due to the similar recurrent structure. The DEV layer training is $\geq 5$ times faster than the continuous-time series baseline NCDE (Kidger et al. (2020)). The training time and memory consumption of LSTM+DEV(SO) on sequential image data are reported in Table 4. The memory of LSTM+DEV(SO) only increases roughly 25% compared to the plain LSTM, with about doubled training time. In comparison to ExpRNN, a geometric RNN model to alleviate gradient issues of RNNs, LSTM+DEV has only half of the model parameters but outperforms consistently, especially in sequential CIFAR-10 data.

### 4.2 Modelling dynamics on non-Euclidean spaces

Brownian motion on $\mathbb{S}^2$ and $N$-body motions are simulated to demonstrate the efficacy of the development network, taking into account the group structure. Our LSTM+DEV model, with Lie groups appropriately chosen, can constrain the prediction to prescribed non-Euclidean space and significantly boost the performance over baseline LSTM models.

### 4.2.1 Brownian motion on $\mathbb{S}^2$

We simulated 20000 samples of the discretised Brownian motion $B$ on the unit 2-sphere $\mathbb{S}^2$ with time length $L = 500$ and equal time spacing $\Delta t = 2e^{-3}$ by the random walk approach (Novikov et al. (2020)). The driving path for simulating $B$ is a random walk $X$ of length $L$ in $\mathbb{R}^2$. The task of predicting the Brownian motion from the given driving random walk is formulated as a sequence-to-sequence supervised learning problem with input $X$ and output $B$. We benchmark our LSTM+DEV model with $SO(3)$-representation

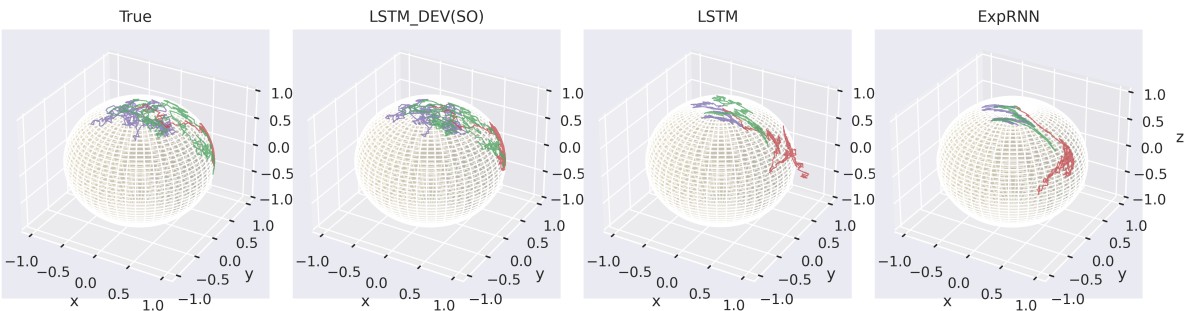

Figure 5: True and predicted sample trajectories of the Brownian motion on $\mathbb{S}^2$ on the test set. The test MSE of LSTM+DEV(SO(3)), LSTM and ExpRNN are **0.0184**, 0.109 and 0.11 respectively.

against the baselines LSTM and ExpRNN. Essentially, the DEV(SO(3)) layer is used as a final pooling layer to map the output trajectories to the constrained Lie group. See Appendix D.5 for details on data simulation and model architecture.

As illustrated in Figure 5, the proposed LSTM+DEV(SO(3)) reduces the test MSE of both baselines by around 83% . Its predicted trajectories always stay on the sphere thanks to the built-in Lie group structure of the development output. It is highlighted that ExpRNN enforces model weights to the orthogonal group, but it can not constrain the output to live in the desired Lie group (see the red sample leaves from the sphere).

Table 5: Test MSE (mean $\pm$ std, computed across 5 runs) on the $N$-body simulation dataset

| | Test MSE (1e-3) | | |
| --- | --- | --- | --- |
| Prediction steps | 100 | 300 | 500 |
| Static | 7.79 | 61.1 | 157 |
| LSTM | 3.54$\pm$ 0.22 | 22.1 $\pm$ 1.5 | 63.2 $\pm$1.4 |
| LSTM+DEV(SO(2)) | 5.50 $\pm$ 0.01 | 47.5$\pm$0.19 | 128$\pm$0.58 |
| LSTM+DEV(SE(2)) | **0.505$\pm$0.01** | **6.94$\pm$ 0.16** | **25.6$\pm$ 1.5** |

### 4.2.2  $N$-body simulations

Following Kipf et al. (2018), we use the simulated dataset of 5 charged particles and consider a regression task of predicting the future $k$-step location of the particles. The input sequence consists of the location and velocity of particles in the past 500 steps. As coordinate transforms from the current location $x_t$ to the future location $x_{t+k}$ lie in $SE(2)$ (the special Euclidean group), we propose to use the LSTM+DEV with $SE(2)$-representation to model such transforms. Details of data simulation and model architecture can be found in Appendix D.6. To benchmark our proposed method, we consider two other baselines: (1) the current location (static) and (2) the LSTM model. To justify the choice of the Lie algebra here, we compare our method with LSTM+DEV(SO(2)).

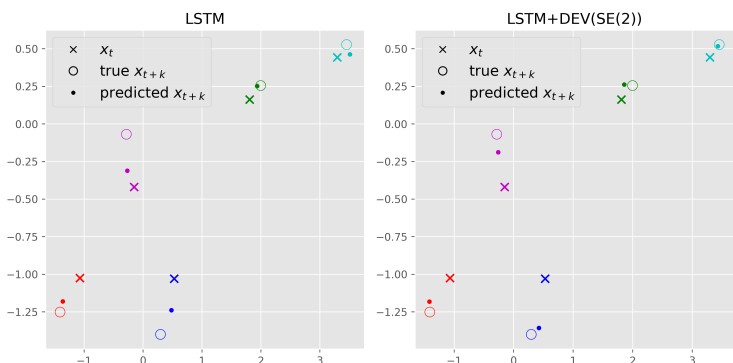

Figure 6: Prediction comparison between LSTM (left) and LSTM+DEV (SE(2)) (right) in the 5-body simulations.

As shown in Table 5, the proposed LSTM+DEV(SE(2)) consistently and significantly outperforms both baselines and LSTM+DEV(SO(2)) in terms of MSE for varying $k \in \{100, 300, 500\}$. Figure 6 provides a visual demonstration to show the superior performance of the estimator of future location by our LSTM+DEV(SE(2)) in comparison to the strongest baseline LSTM, thus manifesting the efficacy of incorporating development layers with inherited group structures in learning tasks.

# 5    Discussions and future work

## 5.1    Vector field

The development layer has a *linear* driving vector field (in terms of controlled differential equations, CDE), which enables us to derive an explicit, analytic solution and use it for fast computation of the development layer. However, this may impose limitation on the development layer as an efficient standalone model for modelling complex time series. In this regard, one may consider extending the development layer by parameterising the vector field via a neural network taking values in the *Lie algebra*, and computing it by solving a nonlinear CDE in analogy with NCDE (Kidger et al. (2020)).

## 5.2    Lie group representation

Although our exposition focuses on matrix groups, our methodology and implementation of the development layer can be naturally extended to general Lie groups. In view of recent works on incorporating Lie group representations to neural networks (Fuchs et al. (2020); Thomas et al. (2018)), we expect that techniques developed therein may be utilised to increase the expressiveness of the development layer and further enhance its performance.

## 5.3    Hybrid model architecture

This work focuses on the proposed hybrid LSTM+Dev model, showing superior performance on various time series learning tasks. However, there is scope to design hybrid models combining the path development layer with other types of neural networks for modelling complex spatio-temporal data, such as video or skeleton-based action sequences. The selection of a suitable hybrid model architecture could potentially enhance performance by leveraging appropriate neural networks (e.g., Convolutional Neural Networks and Graph Neural Networks) to capture spatial dependencies, while utilizing the path development layer to extract temporal dynamics information.

# 6   Conclusion

In this paper, we propose for sequential data tasks a novel, trainable path development layer with finite-dimensional matrix Lie groups built in. Empirical experiments demonstrate that the path development layer, by virtue of its algebraic/geometric structures and recurrent nature, proves advantageous in improving the accuracy and training stability of LSTM models. Based on the path development layer, a compact hybrid LSTM+DEV has been designed, which exhibits competitive performance and superior robustness. It may open up doors to efficient, accurate modelling and prediction of a wide range of time series data with equivariance structures arising from the particular geometries of the underlying domains, such as Hamiltonian dynamical systems, rigid body motion, and molecular dynamics, on various non-Euclidean spaces with nontrivial symmetry groups.

## Acknowledgments

HN is supported by the EPSRC under the program grant EP/S026347/1 and the Alan Turing Institute under the EPSRC grant EP/N510129/1. The research of SL is supported by NSFC Project #12201399, National Natural Science Foundation of China, and Shanghai Frontier Research Institute for Modern Analysis. SL and HN are both supported by the SJTU-UCL joint seed fund WH610160507/067 and the Royal Society International Exchanges 2023 grant (IEC/NSFC/233077). LH is supported by University College London and the China Scholarship Council under the UCL-CSC scholarship (No. 201908060002). The authors extend their gratitude to Terry Lyons for insightful discussions. HN thanks Kevin Schlegel and Jiajie Tao for proofreading the paper. Moreover, HN is grateful to Jing Liu for her help with Figure 1.

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

# A    The signature of a path

In this section, we briefly introduce the signature of a path, which can be used as a principled and efficient feature of time series data Lyons et al. (2007); Levin et al. (2013); Chevyrev & Kormilitzin (2016); Kidger et al. (2019).

Denote by $\mathcal{V}_1\left([0,T], \mathbb{R}^d\right)$ the space of continuous paths $[0,T] \to \mathbb{R}^d$ with finite length. Write $T\left(\left(\mathbb{R}^d\right)\right)$ for tensor algebra space:

$$T\left(\left(\mathbb{R}^d\right)\right) := \bigoplus_{k \geq 0}\left(\mathbb{R}^d\right)^{\otimes k},$$

equipped with tensor product and component-wise addition. The signature takes values in $T\left(\left(\mathbb{R}^d\right)\right)$, and its zeroth component is always 1.

**Definition A.1** (Path Signature). *Let $J \subset [0,T]$ be a compact interval and $X \in \mathcal{V}_1\left([0,T], \mathbb{R}^d\right)$. The signature of $X$ over $J$, denoted by $S(X)_J$, is defined as follows:*

$$S(X)_J = \left(1, \mathbf{X}_J^1, \mathbf{X}_J^2, \cdots\right),$$

*where for each $k \geq 1$, $\mathbf{X}_J^k = \int_{\substack{u_1 < \cdots < u_k \\ u_1, \ldots, u_k \in J}} dX_{u_1} \otimes \cdots \otimes dX_{u_k}$.*

The truncated signature of $X$ of order $k$ is

$$\pi_k(S(X))_J \equiv S^k(X)_J := \left(1, \mathbf{X}_J^1, \mathbf{X}_J^2, \cdots, \mathbf{X}_J^k\right). \tag{8}$$

Its dimension equals $\sum_{i=0}^{k} d^i = \frac{d^{k+1}-1}{d-1}$, which grows exponentially in $k$. The range of signature of all paths in $\mathcal{V}_1\left([0,T], \mathbb{R}^d\right)$ is denoted as $S\left((\mathcal{V}_1([0,T], \mathbb{R}^d)\right)$.

**Example A.1** (Linear paths). *For a linear path $X : [0,T] \to \mathbb{R}^d$, its signature is given by*

$$S(X)_{0,T} = \exp\left(X_T - X_0\right),$$

*where* $\exp$ *is the tensor exponential. Equivalently, the $k^{th}$ level of the signature equals*

$$X_{[0,T]}^k = \frac{1}{k!}(X_T - X_0)^{\otimes k}.$$

**Example A.2** (Piecewise linear paths). *Let $x := (x_0, x_1 \cdots, x_N) \in \mathbb{R}^{d \times N}$ be a discrete d-dimensional time series. Embed it in the piecewise linear path $X$ via linear interpolation. By the multiplicative property, the signature of $X$ is given by*

$$S(X) = \exp(x_1 - x_0) \otimes \exp(x_2 - x_1) \otimes \cdots \exp(x_N - x_{N-1}).$$

For a general continuous path of bounded variation, its signature can be obtained by taking limits of signatures of their piecewise linear approximations as the time mesh tends to 0. For the more general case of paths of bounded $p$-variation $(p \geq 1)$, see Lyons et al. (2007).

The signature can be deemed as a non-commutative version of the exponential map defined on the space of paths. Indeed, consider $\exp : \mathbb{R} \to \mathbb{R}$, $\exp(t) = e^t = \sum_{k=0}^{\infty} \frac{t^k}{k!}$; it is the unique $C^1$-solution to the linear differential equation $d \exp(t) = \exp(t)dt$. Analogously, the signature map $t \mapsto S(X_{0,t})$ solves the following linear differential equation:

$$\begin{cases} dS(X)_{0,t} = S(X)_{0,t} \otimes dX_t, \\ S(X)_{0,0} = \mathbf{1} := (1, 0, 0, \ldots) \end{cases} \tag{9}$$

The signature provides a top-down description of the path. It serves as an attractive tool for machine learning in light of the properties manifested in the three results below.

**Lemma A.1** (Invariance under time reparameterization; Lyons et al. (2007)). *Let $X \in \mathcal{V}^1\left([0,T],\mathbb{R}^d\right)$ and $\lambda : [0,T] \to [T_1, T_2]$ be a non decreasing surjection and define $X_t^\lambda := X_{\lambda_t}$ for the representation of $X$ under $\lambda$. Then for every $s, t \in [0,T]$,*

$$S(X)_{\lambda_s, \lambda_t} = S\left(X^\lambda\right)_{s,t} \tag{10}$$

**Theorem A.1** (Uniqueness of signature; Hambly & Lyons (2010)). *Let $X \in \mathcal{V}^1([0,T], E)$. Then $S(X)$ determines $X$ up to the tree-like equivalence.*

**Theorem A.2** (Signature Universal Approximation; Lyons et al. (2007)). *Suppose $f : S_1 \to \mathbb{R}$ is a continuous function, where $S_1$ is a compact subset of $S\left(\mathcal{V}^p(J, \mathbb{R}^d)\right)$ and $J \subset \mathbb{R}$ is an interval. The topology on $\mathcal{V}^p\left(J, \mathbb{R}^d\right)$ is chosen such that the signature map $S$ is continuous. Then for any $\epsilon > 0$ there is a linear functional $L \in T\left(\left(\mathbb{R}^d\right)\right)^*$ such that*

$$|f(a) - L(a)| \leq \epsilon \qquad \text{for every } a \in S_1.$$

# B   The development of a path

## B.1   Connection with signature

Recall that $\mathcal{V}_1\left([0,T], \mathbb{R}^d\right)$ denote the space of continuous paths $[0,T] \to \mathbb{R}^d$ of bounded 1-variation and $\mathfrak{g}$ is a (matrix) Lie algebra.

**Lemma B.1** (Lemma 2.2). *Let $X \in \mathcal{V}_1\left([0,T], \mathbb{R}^d\right)$ and $M \in \mathbf{L}\left(\mathbb{R}^d, \mathfrak{g}\right)$. The development of the path $X$ under $M$ is given by*

$$D_M(X) = \widetilde{M}\big(S(X)\big).$$

*Proof.* By definition of the development, $D_M(X)$ is the endpoint of $(Z_t)_{t \in [0,T]}$, which satisfies the linear controlled differential equation

$$\mathrm{d}Z_t = Z_t M(\mathrm{d}X_t), \qquad Z_0 = \mathrm{Id}.$$

By Picard iteration, one obtains

$$D_M(X) := Z_T = \mathrm{Id} + \sum_{n \geq 1} \int_{0 < t_1 < \cdots < t_n < T} M(\mathrm{d}X_{t_1}) \cdots M(\mathrm{d}X_{t_n}).$$

Thus, by linearity of $M$, the right-hand side of the above equation coincides with $\widetilde{M}\big(S(X)\big)$.  $\square$

## B.2   Dimension comparison with signature

The dimension of the signature of a $d$-dimensional path of depth $k$ is $\frac{d^{k+1}-1}{d-1}$ if $d \geq 2$ and $k$ if $d = 1$. When specifying the Lie algebra $\mathfrak{g}$ of the development as a matrix algebra, the dimension of development of order $m$ is $m^2$ (independent of $d$). The dimension of trainable weights of the development layer with static output is $dm^2$, while the signature layer does not have any trainable parameter.

## B.3   Invariance under time-reparameterisation

**Lemma B.2** (Lemma 2.3). *Let $X \in \mathcal{V}_1\left([0,T], \mathbb{R}^d\right)$ and let $\lambda$ be a non-decreasing $C^1$-diffeomorphism from $[0, S]$ onto $[0,T]$. Define $X_t^\lambda := X_{\lambda_t}$ for $t \in [0,T]$. Then for all $M \in \mathbf{L}\left(\mathbb{R}^d, \mathfrak{g}\right)$,*

$$D_M(X_{\lambda_s, \lambda_t}) = D_M\left(X_{s,t}^\lambda\right).$$

*Proof.* It follows directly from Lemma 2.2 and the invariance under time reparameterisation of the signature. See Chevyrev & Kormilitzin (2016).  $\square$

### B.4 Universality

Let $W$ be a vector space. In the sequel, we denote by $\mathfrak{gl}(W)$ the space of endomorphisms of $W$ equipped with the Lie bracket $[\sigma_1, \sigma_2] := \sigma_1 \circ \sigma_2 - \sigma_2 \circ \sigma_1$.

**Definition B.1** (Lie algebra representation). *Let $\mathfrak{g}$ be a Lie algebra. A Lie algebra representation of $\mathfrak{g}$ on a vector space $W$ is a Lie algebra homomorphism $\mathfrak{g} \to \mathfrak{gl}(W)$. That is, a linear map satisfying $\rho([X_1, X_2]) = [\rho(X_1), \rho(X_2)]$ for all $X_1, X_2 \in \mathfrak{g}$.*

To fix the idea, endow with $(\mathbb{R}^d)^{\otimes k}$ for each $k \in \mathbb{N}$ the projective norm; see Chevyrev & Lyons (2016). Let $\mathcal{E}$ be the space of tensor elements with infinite radius of convergence:

$$\mathcal{E} := \left\{ x := (x_0, x_1, \cdots) \in T((\mathbb{R}^d)) : \sum_{k=0}^{\infty} \|x_k\| \lambda^k < \infty \text{ for all } \lambda \geq 0 \right\}.$$

**Theorem B.1** (Universality of path development). *Let $\mathcal{G}$ be the space of group-like elements in $\mathcal{E}$ and $\mathcal{K} \subset \mathcal{G}$ be a compact subset. For any continuous function $f : \mathcal{K} \to \mathbb{C}$ and $\epsilon > 0$, there exist $\widetilde{M_1}, \cdots, \widetilde{M_N} \in \mathbf{L}(\mathcal{G}, \mathfrak{gl}(m, \mathbb{C}))$ and $L_1, \ldots, L_N \in \mathbf{L}(\mathfrak{gl}(m, \mathbb{C}); \mathbb{C})$ such that*

$$\sup_{x \in \mathcal{K}} \left| f(x) - \sum_{i=1}^{N} L_i \circ \widetilde{M_i}(x) \right| < \epsilon.$$

Here recall from Eq. (4) that $\widetilde{M_j} \in \mathbf{L}(\mathcal{G}, \mathfrak{gl}(m, \mathbb{C}))$ is the algebra morphism extended from $M_j \in \mathbf{L}(\mathbb{R}^d; \mathfrak{gl}(m, \mathbb{C}))$ by naturality. As per Definition 2.1 in Chevyrev & Lyons (2016), equip $T((\mathbb{R}^d))$ with the coarsest topology such that all algebra morphisms $\widetilde{M} \in \mathbf{L}(\mathcal{G}, \mathcal{A})$ which arise from $M \in \mathbf{L}(\mathbb{R}^d; \mathcal{A})$, with $\mathcal{A}$ ranging through all normed algebras, are continuous.

Also, note that when the algebraic homeomorphisms $\widetilde{M_i}$ have their domains restricted to $\mathcal{G}$, they indeed take values in the group $\mathrm{GL}(m, \mathbb{C})$ of invertible matrices.

*Proof.* Let us consider

$$\mathcal{A} := \left\{ \text{finite-dimensional matrix representations of } T((\mathbb{R}^d)) \right.$$

$$\left. \text{arising from extensions of all } M \in \mathbf{L}\left(\mathbb{R}^d; \bigoplus_{m=1}^{\infty} \mathfrak{gl}(m, \mathbb{C})\right) \right\}.$$

By Chevyrev & Lyons (2016), Theorem 4.8, $\mathcal{A}$ separates points over $T((\mathbb{R}^d))$. Then define

$$\mathcal{B} := \left\{ \lambda \circ \widetilde{M} : \widetilde{M} \in \mathcal{A} \text{ and } \lambda \in \mathbf{L}\left(\bigoplus_{m=1}^{\infty} \mathfrak{gl}(m, \mathbb{C}); \mathbb{C}\right) \right\} \subset \mathbf{L}(\mathbb{R}^d; \mathbb{C}). \tag{11}$$

We *claim* that $\mathcal{B}$ is a complex unital *-algebra.

We first explain how to conclude the proof assuming the *claim*. Indeed, since $\mathcal{A}$ separates points over $T((\mathbb{R}^d))$, we know that $\mathcal{B}$ separates points over $T((\mathbb{R}^d))$ too. By Corollary 2.4 in Chevyrev & Lyons (2016), we know that $\mathcal{G}$ is Hausdorff, so $\mathcal{K} \subset \mathcal{G}$ is a compact Hausdorff space. It thus follows from the Stone–Weierstrass theorem — *i.e.*, for a compact, Hausdorff topological space $\mathcal{X}$, if $S \subset C^0(\mathcal{X}, \mathbb{C})$ separates points, then the complex unital *-algebra generated by $S$ is dense in $C^0(\mathcal{X}, \mathbb{C})$ — that for any $f : \mathcal{K} \to \mathbb{C}$ and $\epsilon > 0$, there exists $S \in \mathcal{B}$ (viewed as a function over the domain $\mathcal{K}$) such that $\|f - S\|_{C^0(\mathcal{K})} < \epsilon/2$. In addition, as both $\mathbf{L}(\mathbb{R}^d; \bigoplus_{m=1}^{\infty} \mathfrak{gl}(m, \mathbb{C}))$ and $\mathbf{L}(\bigoplus_{m=1}^{\infty} \mathfrak{gl}(m, \mathbb{C}); \mathbb{C})$ have countable bases, $\mathcal{B}$ is separable. Thus one may find $\widetilde{M_1}, \cdots, \widetilde{M_N} \in \mathbf{L}(\mathcal{G}, \mathfrak{gl}(m, \mathbb{C}))$ and $L_1, \ldots, L_N \in \mathbf{L}(\mathfrak{gl}(m, \mathbb{C}); \mathbb{C})$ satisfying $\left\| S - \sum_{i=1}^{N} L_i \circ \widetilde{M_i} \right\|_{C^0(\mathcal{K})} < \epsilon/2$. This concludes the proof.

Now let us prove the *claim*.

- $\mathcal{A}$ contains the unit, which is the trivial representation. The unit of $\mathbf{L}\left(\bigoplus_{m=1}^{\infty}\mathfrak{gl}(m,\mathbb{C})\right)$ is the constant map 1. So $\mathcal{B}$ contains the unit too, which is also the constant map 1.

- The product on $\mathcal{A}$ is given by the tensor product, *i.e.*, the Kronecker product for matrices: $\rho_1 \otimes \rho_2(x) := \rho_1(x) \otimes \rho_2(x)$ for $\rho_1, \rho_2 \in \mathcal{A}$. The involution in $\mathcal{A}$ corresponds to the dual representation, namely that $\rho^*(x) := -[\rho(x)]^*$ (the adjoint matrix) for $\rho \in \mathcal{A}$. By the definition of $M \mapsto \widetilde{M}$, we have $\left(\widetilde{M_1} \otimes \widetilde{M_2}\right)(x) = (M_1 \otimes \widetilde{I + I} \otimes M_2)(x)$, where $I$ is the constant map which maps $x$ to the identity of the group.

- Thus, for $\lambda_1, \lambda_2 \in \mathbf{L}\left(\bigoplus_m \mathfrak{gl}(m,\mathbb{C});\mathbb{C}\right)$ and $M_1, M_2 \in \mathbf{L}\left(\mathbb{R}^d; \bigoplus_{m=1}^{\infty}\mathfrak{gl}(m,\mathbb{C})\right)$, we may define the natural product $\bullet$ and involution $\star$ in $\mathcal{B}$ as follows:

$$\left[\lambda_1 \circ \widetilde{M_1}\right]^\star = \overline{\lambda_1 \circ \widetilde{M_1}^*},$$
$$\left(\lambda_1 \circ \widetilde{M_1}\right) \bullet \left(\lambda_2 \circ \widetilde{M_2}\right) = (\lambda_1 \otimes \lambda_2) \circ \left(\widetilde{M_1 \otimes M_2}\right).$$

It is easy to check that they are well defined and satisfy the desired properties as algebraic operations on unital $^*$-algebra over $\mathbb{C}$. In particular, their continuity follows from that of $M \mapsto \widetilde{M}$, which is part of the definition of the topology on $T((\mathbb{R}^d))$.

The *claim* is proved. $\qquad\square$

Two remarks are in order concerning the above theorem.

**Remark B.1.** *1. It may appear odd that the product in $\mathcal{A}$ has the structure of tensor product of* group *representations instead of* Lie algebra *representations. This is intentionally designed to make $\mathcal{A}$ (with domain restricted to $\mathcal{G}$) a* Hopf *algbera representation of the space of group-like elements $\mathcal{G}$. Indeed, one characterisation of $\mathcal{G}$ is that*

$$\mathcal{G} = \left\{x \in T((\mathbb{R}^d)) \setminus \{0\} : \blacktriangle(x) = x \otimes x\right\},$$

*where $\blacktriangle$ is the Hopf algebra coproduct on the tensor algebra, extended from $\blacktriangle(v) = v \otimes \mathbf{1} + \mathbf{1} \otimes v$ on $\mathbb{R}^d$. In this provision $(\rho_1 \otimes \rho_2)(x) \equiv (\rho_1 \otimes \rho_2)[\blacktriangle(x)]$. On the other hand, the sign in the dual representation corresponds to the antipode of the Hopf algebra.*

*2. Our version of $\mathcal{A}$ is taken only for convenience of exposition; it is larger than necessary. In fact, we may adopt verbatim Chevyrev & Lyons (2016), Definition 4.1 for the definition of $\mathcal{A}$, in which only $M$ taking values in unitary operators on finite-dimensional complex Hilbert spaces are taken into account. In this way, we may replace $\mathfrak{gl}(m,\mathbb{C})$ in Theorem B.1 by the unitary Lie algebra $\mathfrak{u}(m)$. The proof is essentially the same, as the Hopf algebra representations of $\mathcal{G}$ into $\bigoplus_m \mathfrak{u}(m)$ remains a complex unital $^*$-algebra separating points over $\mathcal{G}$.*

## C  Backpropagation of the development layer

### C.1  Backpropagation of the development

Consider a scalar field $\psi : G^{N+1} \to \mathbb{R}$ and an input $x := (x_n)_{n=0}^N \in \mathbb{R}^{N+1}$. We aim to find optimal parameters $\theta^*$ minimising $\psi(\mathcal{D}_\theta(x))$; in formula,

$$\theta^* = \mathrm{argmin}_{\theta \in \mathfrak{g}^d} \ \psi(\mathcal{D}_\theta(x)),$$

where $\mathcal{D}_\theta$ is the development layer defined in Eq. (6). Without loss of generality, we focus on the case where the output is the full sequence of path development. Euclidean gradient descent is not directly applicable here due to the Lie group structure of the output.

The following identity allows us to compute the gradient of Lie algebra-valued parameters in our development layer by adapting the method of trivialisation in Lezcano-Casado (2019); Lezcano-Casado & Martınez-Rubio (2019). Consider a smooth map $\Phi : \mathcal{N} \to \mathcal{M}$ between Riemannian manifolds $(\mathcal{M}, g_1)$ and $(\mathcal{N}, g_2)$. Denote its differential by $d\Phi : T\mathcal{N} \to T\mathcal{M}$, and let the adjoint of $d\Phi$ with respect to $g_1, g_2$ be $d\Phi^* : T\mathcal{M} \to T\mathcal{N}$.

**Corollary C.1.** *Let $\Phi : \mathcal{N} \to \mathcal{M}$ be a smooth map between Riemannian manifolds and let $f : \mathcal{M} \to \mathbb{R}$. Then $\nabla(f \circ \Phi) = \mathrm{d}\Phi^*(\nabla f)$.*

### C.2 Proof of Theorem 3.1

*Proof.* We first consider a more general statement: let $\mathcal{M}$ and $\mathcal{N}$ be differentiable manifolds, let $f : \mathcal{M} \to \mathcal{N}$ be a smooth function, and let $\widetilde{\psi} : \mathcal{M} \to \mathbb{R}$ be a scalar field factors through the graph of $f$. That is, there is a smooth function $\psi : \mathcal{M} \times \mathcal{N} \to \mathbb{R}$ such that $\widetilde{\psi}(x) = \psi(x, f(x))$ for each $x \in \mathcal{M}$. Then we have the following identity on $T^*\mathcal{M}$:

$$\mathrm{d}\widetilde{\psi} = (\mathbf{pr}_{\mathrm{I}})_{\#}\,\mathrm{d}\psi + \left[(\mathbf{pr}_{\mathrm{II}})_{\#}\,\mathrm{d}\psi\right] \circ \mathrm{d}f. \tag{12}$$

Here we view $(\mathbf{pr}_{\mathrm{I}})_{\#}\,\mathrm{d}\psi \in T^*\mathcal{M} \oplus \{0\}$ and $\left[(\mathbf{pr}_{\mathrm{II}})_{\#}\,\mathrm{d}\psi\right] \circ \mathrm{d}f \in \{0\} \oplus T^*\mathcal{M}$; the symbols $\mathbf{pr}_{\mathrm{I}}$ and $\mathbf{pr}_{\mathrm{II}}$ denote the canonical projections from $\mathcal{M} \times \mathcal{N}$ onto $\mathcal{M}$ and $\mathcal{N}$, respectively.

The demonstration for Eq. (12) is straightforward. For functions $f_1 : E \to S_1$ and $f_2 : E \to S_2$ where $E, S_1, S_2$ are arbitrary sets, we denote $f_1 \oplus f_2 : E \to (S_1 \times S_2)$ as the function $(f_1 \oplus f_2)(x) := (f_1(x), f_2(x))$. In this notation one has

$$\widetilde{\psi} = \psi \circ (\mathrm{Id}_{\mathcal{M}} \oplus f).$$

Using the chain rule we then deduce that

$$\mathrm{d}\widetilde{\psi} = \mathrm{d}\psi\big|_{\mathrm{range}\,(\mathrm{Id}_{\mathcal{M}} \oplus f)} \circ \mathrm{d}\,(\mathrm{Id}_{\mathcal{M}} \oplus f) = \mathrm{d}\psi\big|_{\mathrm{graph}\,f} \circ (\mathrm{Id}_{T\mathcal{M}} \oplus \mathrm{d}f),$$

with $\mathrm{Id}_{T\mathcal{M}} : T\mathcal{M} \to [T\mathcal{M} \cong T\mathcal{M} \oplus \{0\}]$ and $\mathrm{d}f : T\mathcal{M} \to [T\mathcal{N} \cong \{0\} \oplus T\mathcal{N}]$. This is tantamount to the right-hand side of Eq. (12) by the definition of $\mathbf{pr}_{\mathrm{I}}$ and $\mathbf{pr}_{\mathrm{II}}$.

Let us now prove the lemma from the identity (12). Indeed, by taking $\mathbf{pr}_n$ in place of $\mathbf{pr}_{\mathrm{I}}$ and $f = \mathfrak{S}_{n+1} \oplus (\mathfrak{S}_{n+2} \circ \mathfrak{S}_{n+1}) \oplus \ldots \oplus (\mathfrak{S}_N \circ \ldots \circ \mathfrak{S}_{n+2} \circ \mathfrak{S}_{n+1})$, where $\mathcal{M} = G$ and $\mathcal{N} = G^{N-n}$, we deduce that

$$\begin{aligned}
\mathrm{d}\widetilde{\psi}_n &= (\mathbf{pr}_n)_{\#}\,\mathrm{d}\psi + \left\{(\mathbf{pr}_{n+1})_{\#}\,\mathrm{d}\psi\right\} \cdot \mathrm{d}\mathfrak{S}_{n+1} + \left\{(\mathbf{pr}_{n+2})_{\#}\,\mathrm{d}\psi\right\} \cdot \mathrm{d}\,(\mathfrak{S}_{n+2} \circ \mathfrak{S}_{n+1}) \\
&\quad + \left\{(\mathbf{pr}_N)_{\#}\,\mathrm{d}\psi\right\} \cdot \mathrm{d}\,(\mathfrak{S}_N \circ \cdots \circ \mathfrak{S}_{n+2} \circ \mathfrak{S}_{n+1}) \\
&= (\mathbf{pr}_n)_{\#}\,\mathrm{d}\psi + \left\{(\mathbf{pr}_{n+1})_{\#}\,\mathrm{d}\psi\right\} \cdot \mathrm{d}\mathfrak{S}_{n+1} + \left\{(\mathbf{pr}_{n+2})_{\#}\,\mathrm{d}\psi\right\} \cdot \mathrm{d}\mathfrak{S}_{n+2} \cdot \mathrm{d}\mathfrak{S}_{n+1} \\
&\quad + \left\{(\mathbf{pr}_N)_{\#}\,\mathrm{d}\psi\right\} \cdot \mathrm{d}\,(\mathfrak{S}_N \circ \mathfrak{S}_{N-1} \circ \cdots \circ \mathfrak{S}_{n+2}) \cdot \mathrm{d}\mathfrak{S}_{n+1}.
\end{aligned}$$

Here we have used the fact that for $\mathcal{L}_g : G \to G$, $\mathcal{L}_g(h) := gh$, its differential $\mathrm{d}\mathcal{L}_g$ is also the left-multiplication by $g$, which is given by matrix multiplication when $\mathfrak{g}$ is a matrix algebra. The second equality follows from the chain rule.

On the other hand, notice that the terms in the right-most expression above possess a recursive structure too: by taking $\mathbf{pr}_{n+1}$ in place of $\mathbf{pr}_{\mathrm{I}}$ and $f = \mathfrak{S}_{n+2} \oplus (\mathfrak{S}_{n+3} \circ \mathfrak{S}_{n+2}) \oplus \ldots \oplus (\mathfrak{S}_N \circ \ldots \circ \mathfrak{S}_{n+2})$ in Eq. (12), we have

$$(\mathbf{pr}_{n+1})_{\#}\,\mathrm{d}\psi + \left\{(\mathbf{pr}_{n+2})_{\#}\,\mathrm{d}\psi\right\} \cdot \mathrm{d}\mathfrak{S}_{n+2} + \left\{(\mathbf{pr}_N)_{\#}\,\mathrm{d}\psi\right\} \cdot \mathrm{d}\,(\mathfrak{S}_N \circ \mathfrak{S}_{N-1} \circ \cdots \circ \mathfrak{S}_{n+2}) = \mathrm{d}\widetilde{\psi}_{n+1}.$$

The above computations establish the following equality on $T^*G$:

$$\mathrm{d}\widetilde{\psi}_n = (\mathbf{pr}_n)_{\#}\,\mathrm{d}\psi + \mathrm{d}\widetilde{\psi}_{n+1} \cdot \mathrm{d}\mathfrak{S}_{n+1}.$$

Thus, evaluating this identity at $z_n \in G$, we infer from the definition of the update function $\mathfrak{S}_{n+1}(z_n) = z_{n+1}$ that

$$\mathrm{d}_{z_n}\widetilde{\psi}_n = (\mathbf{pr}_n)_{\#}\,\mathrm{d}\psi\big|_{z_n} + \mathrm{d}_{z_{n+1}}\widetilde{\psi}_{n+1} \cdot \mathrm{d}_{z_n}\mathfrak{S}_{n+1}.$$

We now conclude by the recurrence relation $z_{n+1} = z_n \cdot \exp(M_{\theta_{n+1}}(\Delta x_{n+1}))$ in Eq. (6). $\qquad\square$

A remark on notations is in order: In the last line of the proof above, we (formally) replaced $\mathrm{d}_{z_n}\mathfrak{S}_{n+1}$ with $\exp(M_{\theta_{n+1}}(\Delta x_{n+1}))$. The reason is that

$$\mathfrak{S}_{n+1}(z_n) = z_{n+1} = z_n \cdot \exp(M_{\theta_{n+1}}(\Delta x_{n+1})),$$

where $\mathfrak{S}_{n+1} : G \to G$. Viewing $G$ as a subset of vector space $\mathfrak{gl}(m, \mathbb{C})$, we may regard $\mathfrak{S}_{n+1}$ as a linear mapping in its argument. Thus, $\mathrm{d}_{z_n}\mathfrak{S}_{n+1} : T_{z_n}G \to T_{z_{n+1}}G$ equals the *right multiplication* by the matrix $\exp\left(M_{\theta_{n+1}}(\Delta x_{n+1})\right)$.

## C.3   Proof of Proposition 3.1

Before proceding to the proof of Proposition 3.1, we remark that the Riemannian metric taken here is the one given by the Hilbert–Schmidt norm of matrices in $\mathcal{M} = \mathfrak{gl}(m, \mathbb{F})$ for $\mathbb{F} = \mathbb{R}$ or $\mathbb{C}$. Equivalently, we identify $\mathfrak{gl}(m, \mathbb{F})$ with the Euclidean space $\mathbb{F}^{m \times m}$ via the diffeomorphism $\Psi : \mathfrak{gl}(m, \mathbb{F}) \to \mathbb{F}^{m \times m}$,

$$\Psi\left(\{A_j^i\}_{1 \le i,j \le m}\right) := \left[A_1^1, A_2^1, \ldots, A_m^1, A_1^2, A_2^2, \ldots, A_m^2, \cdots\cdots, A_1^m, A_2^m, \ldots, A_m^m\right]^\top.$$

The pullback metric is precisely the Hilbert–Schmidt metric $\langle B, C \rangle = \mathrm{tr}(BC^*)$ for any $B, C \in T_A\mathfrak{gl}(m, \mathbb{F}) \cong \mathfrak{gl}(m, \mathbb{F})$ with arbitrary $A \in \mathfrak{gl}(m, \mathbb{F})$.

The choice of Hilbert–Schmidt metric on matrix Lie groups is natural in our context. Indeed, for inclusions $SO(m) \subset \mathfrak{gl}(m, \mathbb{R})$ and $U(m) \subset \mathfrak{gl}(m, \mathbb{C})$, pullback metrics of the Hilbert–Schmidt metric under the natural inclusions are exactly the bi-invariant Riemannian metrics on $SO(m)$ and $U(m)$, respectively.

In provision of this remark, one can compute the gradient of $\widetilde{\psi}_n$ in Eq. (7) by expressing $\nabla_\theta \widetilde{\psi}_n$ in terms of the differential of exponential map $\exp : \mathfrak{g} \to G$, which in turn is computed via Theorem 2.2 and Lemma 2.4. Recall the projection $\mathbf{Proj} : \mathfrak{gl}(m, \mathbb{C}) \to \mathfrak{g}$ from Eq. (5).

As usual, denote by $\mathcal{L}_{g_1} : G \to G$ for each $g_1 \in G$ the left regular action; *i.e.*, $\mathcal{L}_{g_1}(g_2) := g_1 g_2$.

*Proof.* The expression

$$\nabla_\theta\left(\psi \circ \mathcal{D}_\theta(x)\right) = \sum_{n=1}^{N} \nabla_\theta\left(\widetilde{\psi}_n \circ \mathfrak{S}_n(z_{n-1}, \theta)\right)$$

follows directly from the definition of $\widetilde{\psi}_n$ in Eq. (7).

To further compute the right-hand side, notice that $\nabla_\theta\left(\widetilde{\psi}_n \circ \mathfrak{S}_n(z_{n-1}, \theta)\right)$ is a vector field on the *Euclidean* manifold $\mathfrak{g}^d$ (see Remark 3.3). Recall also that

$$z_n = \mathfrak{S}_n(z_{n-1}, \theta) = \mathcal{L}_{z_{n-1}}\left[\exp\left(M_\theta(\Delta x_n)\right)\right],$$

where $M_\theta(\Delta x_n)$ takes values in $\mathfrak{g} = \mathbf{Proj}(\mathfrak{gl}(m, \mathbb{C}))$. Moreover, set $A : \mathfrak{g}^d \to \mathfrak{g}$ to be the mapping $A(\theta) := M_\theta(\Delta x_n)$ and put $f = \widetilde{\psi}_n \circ \mathcal{L}_{z_{n-1}}$. We compute that

$$\nabla_\theta\left(\widetilde{\psi}_n \circ \mathfrak{S}_n(z_{n-1}, \theta)\right) = \mathbf{Proj}\left\{A^\# \nabla(f \circ \exp)\left(A(\theta)\right)\right\}$$

$$= \mathbf{Proj}\left\{A^\#\left\{\mathrm{d}_{A(\theta)^\top}\left[\nabla f(\exp\left(A(\theta)\right))\right]\right\}\right\}.$$

The second line follows from Theorem 2.2, and the first line holds by the general rule — let $\mathcal{M}$ and $\mathcal{N}$ be Riemannian manifolds and $\varphi : \mathcal{M} \to \mathcal{N}$, $h : \mathcal{N} \to \mathbb{R}$ be smooth functions. Then $\nabla(h \circ \varphi) = \varphi^\#(\nabla h)$ where $\varphi^\#$ is the pullback operator.

To proceed, let us compute $\nabla f$. Since for any $g \in G$ the left regular action $\mathcal{L}_g$ satisfies $\mathrm{d}\mathcal{L}_g = \mathcal{L}_g$ *whenever $G$ is a matrix group*, we have that

$$\nabla f = \nabla_{z_n}\widetilde{\psi}_n \cdot z_{n-1},$$

where $\cdot$ is the matrix multiplication. We put the subscript $z_n$ here only to emphasise that the gradient of $\widetilde{\psi}_n$ is taken with respect to this variable.

It remains to compute $(A^{\#}V)(\theta)$, where we write $V := \mathrm{d}_{A(\theta)^{\top}}\left[\nabla f(\exp\left(A(\theta)\right)\right]$ for abbreviation. Recall that $V$ and $A^{\#}V$ are vector fields on $\mathfrak{g}$ and $\mathfrak{g}^d$, respectively. Elementary computations in differential geometry then yield that

$$A^{\#}V(\theta) = V\left(A_{\#}\theta\right) = V\left(A(\theta)\right) \qquad \text{for each } \theta \in \mathfrak{g}^d.$$

It is crucial here that $A$ is a *linear* mapping from $\mathfrak{g}^d$ to $\mathfrak{g}$; thus, the pushforward $A_{\#}(\theta)$ coincides with $A(\theta)$.

We may now complete the proof. $\qquad\qquad\square$

## D Experimental details

### D.1 General remarks

**Optimisers.** All experiments used the ADAM optimiser as in Kingma & Ba (2014). Learning rates and batch sizes vary along experiments and models. The learning rate in certain experiments may be applied with a constant exponential decay rate, and the training process is terminated when the metric failed to improve for some large number of epochs. See individual sections for details. We saved model checkpoints after every epoch whenever the validation set performance (the performance metric varies from experiment to experiment) gets improved and loaded the best performing model for evaluation on test set.

**Architectures.** In our numerical experiments, we consider the development models (DEVs) and the hybrid model (LSTM+DEVs) by stacking the LSTM with the development layer together. To benchmark our development-based models, we use the signature model and the LSTM model as two baselines.

For the Speech Commands and Character Trajectories experiments, all the above mentioned models (DEVs, LSTM+DEVs, Signature, and LSTM) all include a linear output layer to predict the estimated probability of each class. For a fair comparison, we keep the total number of parameters across different models comparable and then compare the model performance in terms of accuracy and training stability.

Similarly, we also keep the model complexity of the baseline LSTM and our proposed hybrid model (LSTM+DEVSE(2)) comparable in the $N$-body simulation task. See the specific architecture of each individual model for those three datasets in Appendix D.2, Appendix D.3, and Appendix D.6, respectively.

**Hyperparameter tunning.** Once our model architecture is fixed for fair comparison, hyperparameter tunning focuses mainly on learning rate and batch size. Specifically, the learning rate is first set to 0.003 and reduces until the good performance is achieved. Then we search the batch size with a grid of $[32, 64, 128]$. The decay factor of learning rate was applied to those models with severe overfitting observed on the validation set. Throughout the experiments, we notice that the hyperparameters have minimal effects on the development network model, while the RNN-based models are very sensitive to hyperparameters.

**Loss.** We used cross-entropy loss applied to the softmax function of the output of the model for the multi-classes classification problem. We used mean squared error loss for the regression problem in the $N$-Body simulation.

**Computing infrastructure.** All experiments were run on five Quadro RTX 8000 GPUs. We ran all models with PyTorch 1.9.1 Paszke et al. (2019) and performed hyperparameter tunning with Wandb Biewald (2020). **Codes.** The codes for reproducing all experiments are included in supplementary material.

### D.2 Speech Commands

We follow the data generating procedure in Kidger et al. (2020) and took a 70%/15%/15% train/validation/test split. The batch size was 128 for every model. Dev(SO), LSTM+Dev(SO) and signature models used a constant learning rate 0.001 throughout. LSTM was trained with learning rate 0.001 and exponential decay rate 0.997. Training terminates if the validation accuracy stops improving for 50 epochs. Set maximum training epochs to 150.

**Signature.** We apply the signature transform up to depth 3 on the input, which converts the input time series to a vector of size 8420 before passing to a output linear layer. The model has 84210 parameters in total.

**LSTM.** The input of the model is passed to a single layer of LSTM (135 hidden units). The output of the LSTM (last time step) is passed to a output linear layer. It has 86140 parameters in total.

**DEV(SO).** The input of the model is passed to the special orthogonal development lyear (54 by 54 matrix hidden units). Then the output of the development (final time step) is passed to a final linear output layer. It has 87490 parameters in total.

**LSTM-DEV(SO).** The input of the model is passed to a single LSTM layer (56 hidden units). Then we pass the output of the LSTM (full sequence) to the special orthogonal development layer (32 by 32 matrix hidden units). Then the output of the development (final time step) is passed to a final linear output layer. It has 85066 parameters in total.

### D.3 Character Trajectories

We follow the approach in Kidger et al. (2020), in which we combined the train/test split of the original dataset and then took a 70%/15%/15% train/validation/test split.

The batch size used was 32 for every model. Dev(SO), LSTM+Dev(SO) and signature model used a constant learning rate of 0.001 throughout the training. LSTM was trained with a learning rate of 0.003, with an exponential decay rate of 0.997. If the validation accuracy stops improving for 50 epochs, we terminate the training process. The maximum training epochs is set to be 150.

**Signature.** We apply the signature transform up to depth 4 on the input, which converts the input time series to a vector of size 340, then it is passed to a linear output layer. The model has 6820 parameters in total.

**LSTM.** The input of the model is passed to a single layer of LSTM (40 hidden units). The output of the LSTM (last time step) is passed to a linear output layer. The model has 8180 parameters in total.

**ExpRNN.** The input of the model is passed to a single layer of ExpRNN (78 hidden units). The output of the ExpRNN (last time step) is passed to a linear output layer. The model has 8054 parameters in total.

**DEV(SO).** The model's input is passed to the special orthogonal development layer (20 by 20 matrix hidden units). Then the output of the development (final time step) is passed to a final linear output layer. The model has 7760 parameters in total.

**LSTM-DEV(SO).** The model's input is passed to a single LSTM layer(14 hidden units). Then we pass the output of the LSTM (full sequence) to the special orthogonal development layer (14 by 14 matrix hidden units). Then the output of the development (final time step) is passed to a final linear output layer. The model has 8084 parameters in total.

### D.4 Sequential MNIST and CIFAR10

The MNIST dataset is a large collection of handwritten digits, with a training set of 60,000 examples and a test set of 10,000 examples. Each $28 \times 28$ pixel valued image is flattened to a sequence of length 784. The permuted task (p-MNIST) applies the same random permutation on the sequence of size 784. The CIFAR10 dataset consists of 60000 (50000 training + 10000 test samples) 32 by 32 colour images with 10 classes. Like the sequential MNIST, the CIFAR10 task flattens the images to pixel by pixel sequences of length 1024.

The batch size used was 128 for both sequential MNIST and CIFAR10. LSTM+DEV(SO) was trained with an inital learning rate of 0.002 and an exponential decay rate of 0.997. We terminate the training process if the validation accuracy stops improving for 50 epochs. The maximum training epochs is set to be 200.

**LSTM+DEV(SO).** A common model architecture is used for both datasets. The input is passed to a single LSTM layer (120 hidden units); the output of the LSTM (full sequence) is passed to the special orthogonal development layer (10 by 10 matrix hidden units). Then, the output of the development (final time step) is

passed to a final linear output layer. The model has 72050 and 73010 parameters for sequential MNIST and CIFAR10, respectively.

### D.5 Brownian motion on the unit 2-sphere $\mathbb{S}^2$

We simulated 20000 samples of discretised Brownian motion on $\mathbb{S}^2$ with time length $L = 500$ and equal time spacing $\Delta t = 2e^{-3}$ by the random walk approach (Novikov et al. (2020)).

The discretised Brownian motion $B = (B_{t_n})_{n=0}^{L-1}$ with $t_n = n\Delta t$ on $\mathbb{S}^2$ is simulated by a driving random walk $X = (X_{t_n})_{n=0}^{L-1}$ on $\mathbb{R}^2$. Let $B_{t_0} = (0, 0, 1)^\top$. We first simulate a 2D random walk $X$ by i.i.d. increments $\Delta X_n := X_{n+1} - X_n$, uniformly distributed $\sim U(-0.5, 0.5)$. Then rescale $X$ by the factor $\sqrt{12\Delta t}$ to approximate $\mathbb{S}^2$-valued Brownian motion over the time interval $\Delta t$. For a generic point $b = (b^{(1)}, b^{(2)}, b^{(3)})^\top \in \mathbb{S}^2$ we use the following basis for $T_b\mathbb{S}^2 \subset \mathbb{R}^3$, the tangent plane of $\mathbb{S}^2$ at $b$:

$$
e_b^1 = \frac{1}{\sqrt{(b^{(1)})^2 + (b^{(3)})^2}} \begin{pmatrix} b^3 \\ 0 \\ -b^1 \end{pmatrix}, \qquad e_b^2 = \frac{1}{\sqrt{(b^{(1)})^2 + (b^{(3)})^2}} \begin{pmatrix} -b^{(1)}b^{(2)} \\ -\left[(b^{(1)})^2 + (b^{(3)})^2\right] \\ b^{(2)}b^{(3)} \end{pmatrix}.
$$

At each step $n \in \{1, \cdots, L-1\}$, we perform standard random walk on the tangent plane orthogonal to the orientation vector of $B_{t_n}$ and project $B_{t_n} + \sqrt{12\Delta t}\Delta X_n$ to $B_{t_n}$ as described in Novikov et al. (2020). In this manner we simulate, by Monte-Carlo, independent samples of the pair of driving random walk and corresponding Brownian motion on $\mathbb{S}^2$.

The simulated input/output trajectories are split into train/validation/test with ratio 80%/10%/10%. All models were trained with learning rate 0.003 and exponential decay rate 0.998. The batch size was 64 for every model. Training was terminated if the validation MSE stopped lowering for 100 epochs. The maximum number of epochs was set to be 300.

For all models, we passed the input through two dense layers (32 hidden units), followed by a single LSTM/orthogonalRNN layer (64 hidden units).

**LSTM.** For the LSTM model, we pass the LSTM output (full sequence) through one dense layer (64 hidden units) and a single linear output layer.

**ExpRNN.** For the ExpRNN model, we pass the ExpRNN output (full sequence) through one dense layer (64 hidden units) and a single linear output layer.

**LSTM+DEV(SO(3)).** we pass the LSTM output (full sequence) to a SO(3) development layers (each has 3 by 3 matrix hidden units). The output (full sequence) of the development layer is a sequence Lie group element in SO(3). Then we take the last column of the SO(3) matrix as the final sequential output.

### D.6 $N$-body simulations

We followed Kipf et al. (2018) to simulate 2-dimensional trajectories of the five charged, interacting particles. The particles carry positive and negative charges, sampled with uniform probabilities, interacting according to the relative location and charges. We simulated 1000 training trajectories, 500 validation trajectories and 1000 test trajectories, each having 5000 time steps. Instead of inferring the dynamics of the complete trajectories as in Kipf et al. (2018), we consider it a regression problem with sequential input: the input data is a sequence of locations and velocities in the past 500 time steps (downsampled to the length of 50). The output is the particle's positions in the future $k$ time steps for $k \in \{100, 300, 500\}$.

More specifically, let $x_t^{(i)}$ and $v_t^{(i)}$ be the 2-D location and velocity of the $i^{\text{th}}$ particle at time $t$. The input and output of the regression problem are $\mathbf{x}_t = \left(\left((x_s^{(i)}, v_s^{(i)})\right)_{s=t-500}^{t}\right)_{i=1}^{5}$ and $\left(x_{t+k}^{(i)}\right)_{i=1}^{5}$, respectively. In 2-D the coordinate transform is described by the special Euclidean group $SE(2)$, which consists of rotations

and translations:

$$\begin{pmatrix} x' \\ y' \\ 1 \end{pmatrix} = T \begin{pmatrix} x \\ y \\ 1 \end{pmatrix} := \begin{pmatrix} R & \begin{matrix} t_x \\ t_y \end{matrix} \\ 0 \quad 0 & 1 \end{pmatrix} \begin{pmatrix} x \\ y \\ 1 \end{pmatrix}, \tag{13}$$

where $T \in \mathrm{SE}(2)$ and $R$ is a 2-D rotation matrix.

The proposed LSTM+DEV(SE(2)) network architecture models the map $x_t^{(i)} \mapsto x_{t+k}^{(i)}$, hence predicting the future location by multiplying the output of the development layer with $x_t^{(i)}$. The model weights change over development layers for different particles.

The other three baselines include (1) the current location $x_t$ (static estimator, assuming no further movements of the particle); (2) the LSTM; and (3) the SO(2) development layer.

All models are trained with a learning rate of 0.001 with a 0.997 exponential decay rate. The batch size used was 128 for every model. The training was terminated if the validation MSE stopped lowering for 50 epochs. The maximum number of epochs was set to be 200.

For all models, we passed the input through two dense layers (16 hidden units), followed by a single LSTM layer (32 hidden units).

**LSTM.** For the LSTM model, we pass the LSTM output (last time step) through one dense layer (32 hidden units) and a single linear output layer.

**LSTM+DEV(SE(2)).** We pass the LSTM output (full sequence) to the five independent SE(2) development layers (each has 3 by 3 matrix hidden units). The output of each independent development layer (final time step) is a Lie group element in SE(2). We apply the learned SE(2) element to the last observed location of each particle to estimate the future locations of the five particles, according to Equation (13).

**LSTM+DEV(SO(2)).** We pass the LSTM output (full sequence) to the five independent SO(2) development layers (each has 2 by 2 matrix hidden units). The output of each independent development layer (final time step) is a Lie group element in SO(2). We multiply the learned SO(2) element with the last observed location of each particle to estimate the future locations of the five particles.

# E   Supplementary numerical results

Here we report the supplementary numerical results on speech Commands dataset (Table 6).

Table 6: Test accuracies of the linear model on development and signature baselines on Speech Commands dataset.

| SIGNATURE | | |
|---|---|---|
| DEPTH $(n)$ | TEST ACCURACY | # FEATURE |
| 1 | 12.3% | 20 |
| 2 | 75.4% | 420 |
| 3 | 85.7% | 8420 |
| 4 | 88.9% | 168420 |
| DEVELOPMENT (SO) | | |
| ORDER $(m)$ | TEST ACCURACY | # FEATURE |
| 5 | 70.5% | 25 |
| 10 | 81.3% | 100 |
| 20 | 84.6% | 400 |
| 30 | 86.2% | 900 |
| 50 | 87.5% | 2500 |
| 100 | 89.0 % | 10000 |
| DEVELOPMENT (SP) | | |
| ORDER $(m)$ | TEST ACCURACY | # FEATURE |
| 6 | 79.1% | 36 |
| 10 | 83.7% | 100 |
| 20 | 85.5% | 400 |
| 30 | 87.1% | 900 |
| 50 | 88.5% | 2500 |
| 100 | 86.3% | 10000 |

