# OpenReview forum: "Path Development Network with Finite-dimensional Lie Group"
_TMLR — Accepted by TMLR_

### Review · Reviewer_BQ81 · 2024-04-12

**Summary Of Contributions:**

This paper proposes a novel lie-group based approach to learning the path signature of data. This is applied to time-series data.

**Audience:**

Yes

**Claims And Evidence:**

Yes

**Requested Changes:**

I would like a comparison with https://arxiv.org/abs/2203.05483, as the methods seem broadly related (ie RNN + Lie Group optimization).

I would also emphasize simplifying the presentation and filling in harder to access information.

**Strengths And Weaknesses:**

Strengths
-----------
* The paper is technically correct.
* The paper's subject of time series data is very important and has a wide audience range.

Weaknesses
--------------
* While technically correct, the actual contribution is quite simple. In particular, it is just a lie group parameterized RNN.
* Related to the above, I also think the presentation is far too involved/obfuscated. I would like for there to be a simpler presentation of the background, and the method could be simplified as well.
* Related to the above, but it would likely be much more accessible if the paper was presented with rough path theory as the motivation (with the actual contribution being framed closer to the RNN perspective) than the rough path theory motivation. Notably, the rough path stuff feels a bit superfluous.
* The experimental results are not very impressive.
* There is a slight breach of anonymity, as the conclusion includes named information.

---

> ### Author Response · Authors · 2024-06-27
>
> **Reply to the weakness section:**
>
> - Although the path development network is related to the Lie group and RNN, we want to highlight that it is not the Lie group parameterization RNN in the literature. These Lie group parameterization RNNs are typically motivated by the following RNN:
> \begin{equation*}
>     h_{t+1} = \sigma (W_t h_{t} + A_{t} X_t),
> \end{equation*}
> where $\sigma$ is the activation function, and $W_t \in \mathbb{R}^{h \times h}$, $h$ is the dimension of the hidden neurons $h_t$. In comparison, our development layer defined as $z_{t+1} = z_{t}\exp(M_{\theta}(X_{t+1} - X_{t}))$.
>
> There are several significant differences between our path development layer and these Lie group parameterization RNNs (abbreviated as LG-RNNs) such as [1] proposed by the reviewer, https://arxiv.org/abs/2203.05483).
>
> (1)  In LG-RNNs, the Lie group structure of  is imposed on the weight matrix $W_t$ (model parameters), whereas the output might not be in the Lie group in general. In contrast, the path development has the Lie group valued output while the model parameters live in the space $g^d$.
>
> (2) The path development only utilizes the matrix exponential and does not use any other activation function $\sigma$. It is also important to note that matrix exponential differs from conventional activation functions (e.g., ReLu or sigmoid). Unlike these element-wise operations, the matrix exponential applies specifically to square matrices and cannot be used in the same manner as typical activation functions.
>
> (3) LG-RNNs could be regarded as RNNs with the weights in the constraint space. However, the path development is a completely separate layer that may be followed by the RNN network architecture (or any other time series model architecture) without restricting the search space of weights in the original model.
>
> Apart from the above discussion, we would like to highlight that we include ExpRNNs as the representative baseline of these Lie group parameterization RNNs in our numerical examples. [1] could be viewed as the low-rank approximation of ExpRNNs. We included [1] as the reference of RNNs with weights restricting to the unitary group in the related work section in the revised paper.
>
> - Contribution and presentation
>
> First, we have simplified the mathematical presentations of Section 3. Please see our response to the minor comments section (ii) by Kb6t for further details.
>
> Second, we keep the presentation of our work from the rough path theory perspective. We hope that our previous reply has clarified the difference between the path development and existing Lie group parameterization RNN. In light of this, we disagree that the rough path theory is superfluous. It is important to note that rough path theory is essential for building the theoretical justification of the path development. For example, one can show the characteristic property and universality for the unitary group case, which are directly relevant to machine learning algorithms. Besides, the perspective of rough path provides a natural link between the path development layer and the continuous time series models (e.g., Neural CDEs), which serves as the theoretical justification for the robustness of the path development to missing data.
>
> - We apologize for this oversight and have removed the acknowledgement section.
>
> - Experimental results: We would like to draw your attention to the variety of our experiments, ranging from standard time series benchmarks to dynamics on the manifold, which is acknowledged by Reviewer Kb6t
> . Besides, we have carried out extensive benchmarking and comparisons against state-of-the-art methods, evaluating performance based on various metrics including accuracy, robustness, training time, and model complexity. Our numerical results show the hybrid model (LSTM+DEV) consistently outperforms the baselines and achieves competitive results as state-of-the-art models, while our model has a much less number of model parameters with superior robustness. This strength of the experiment section has been recognized by Reviewer cmMa, who commented, "The experimental section is strong."
>
>
> **Reply to requested change section:**
>
> - Please refer to the first point of our reply to the weakness section for details.
>
> - Please see the "Contribution and presentation" section of our reply to the weakness section.
>
> Reference
>
> [1] projUNN: efficient method for training deep networks with unitary matrices, Kiani, B., Balestriero, R., LeCun, Y. and Lloyd, S., 2022.  Advances in Neural Information Processing Systems, 35, pp.14448-14463.

---

### Review · Reviewer_cmMa · 2024-04-30

**Summary Of Contributions:**

This paper proposes a new method for path development using finite dimensional Lie group algebra.

The idea of the paper is to learn hidden states that conform to a dynamical system inspired behavior, while using an RNN like mechanism, while providing theoretical understanding and guarantees regarding the behavior of the network, as well as allowing to operate on non-Euclidean spaces.

The authors verify their method by numerous experiments, showing the strong performance of the method.

**Audience:**

Yes

**Broader Impact Concerns:**

No concerns.

**Claims And Evidence:**

Yes

**Requested Changes:**

Please see in my review.

**Strengths And Weaknesses:**

Strengths:
- The method proposed in this paper looks novel to me.
- The theory seems correct and provides understanding about the method
- The experimental section is strong

Weaknesses:
- Missing references and qualitative comparisons with literature in the field of neuroODEs. For example see [1] which is a prior work to Chen et al. (neuroODEs) and [2,3] that discuss weight sharing for learning hidden states similar to the paper here. A discussion should be added to the revised paper.

- The paragraph "The preservation of favourable geometric and analytic properties makes the path development a promising
feature representation of sequential data. Specifically, non-commutativity of the multiplication in ....
reflects the irreversibility of the order of events. It has been established in Chevyrev & Lyons (2016) that,
with g suitably chosen, the path development constitutes universal and characteristic features. Moreover, in
contrast to the infinite dimensionality of the path signature, the development takes values in finite-dimensional
Lie groups with dimensions independent of the path dimension d."  is vague and should be expanded.

- The method uses matrix exponentials, which is an expensive thing to compute. How do you cope with it?

- In the key advantages: What does characteristic mean? do you mean universal in approximation sense?

- The authors do not discuss the complexity and runtimes of the methods. It should be added to the paper including a comparison with other methods.

Questions:
- From your notations, in the end of page 2, it looks like there is only time dimension. Does your work support spatial data in addition to temporal data? If so, how? If not, can you elaborate on why?

Minor:
- Typo in the sentence "Consider a matrix Lie group G. It
will shall serve as the range for the development."

- Typo in "it can model the dynamics on on-Euclidean spaces by exploiting appropriate Lie group structure." ?

- Table 1 should be resized, its font looks weird.


References:

[1] Stable Architectures for Deep Neural Networks

[2] Anti-Symmetric DGN: a stable architecture for Deep Graph Networks

[3] Feature Transportation Improves Graph Neural Networks

---

> ### Author Response · Authors · 2024-06-27
>
> - **References and qualitative comparisons with literature in the field of neuroODEs.**
>
> The proposed development layer, defined as the solution to the controlled differential equation driven by a BV-path $X: [0, T] \rightarrow \mathbb{R}^{d}$, is specifically designed for time series data. The most suitable Neural ODE counterpart, *NeuralCDE*, has been chosen in our paper as the baseline. Its comparison with path development layer is detailed in Sections 1.1 & 4 from both the qualitative and quantitative perspectives, respectively.
>
> The neural networks in [1--3] are all motivated by the ODE:
> $$
> \dot{y_t} = \sigma\big(W(t)y(t) + b(t)\big),
> $$
> with $y: [0, T] \rightarrow \mathbb{R}^d$, weight matrix $W(t) \in \mathbb{R}^{d \times d}$, and  activation function $\sigma$. Nonetheless, these models are distinctively different from the path development. First, path development satisfies the *linear ODE*
> $\dot{y_t} = y_t \cdot M_{\theta}(y_t)$ , which involves no activation function at all. Second, the output of path development lives in matrix groups, in stark contrast to [1--3]. Third, the development uses a linear function $M: \mathbb{R}^d \rightarrow \mathfrak{g}$ for a flexibly chosen Lie algebra $\mathfrak{g}$, while in [2, Eq.(2)] the weight $W_t$ between nodes lives in $\mathbb{R}^{d \times d}$ and is not related to the Lie algebra. Finally, we added [1] to the discussions on NODEs in the related work section; refs. [2,3] proposed by the reviewer pertains to networks designed specifically for graph data rather than time series data, hence falling out of scope for our selection of baselines.
>
> - **Refining the introduction:**
>
> We changed the sentences "Roughly speaking ... many references cited therein" to
> "Roughly speaking, the signature of a path serves as a principled feature, offering a top-down description of the path. Just as the monomial basis of $\mathbb{R}^d$, the signature --- viewed as their noncommutative analogue --- constitutes a basis for the path space. More specifically, the signature of $X$ is defined as an infinite sequence $\left(1, \mathbf{X_{[0,T]}^{(1)}}, \cdots,  \mathbf{X_{[0,T]}^{(k)}}, \cdots \right)$, where $$\mathbf{X_{[0, T]}^{k}} = \int_{0 < t_{1} < \cdots t_k<  T} d X_{t_1} \otimes \cdots \otimes d X_{t_k},$$ provided that the above integral is well defined. We refer readers to Section 2.1 and Appendix A for the precise definition of the signature of paths of bounded variation. For a more general case of paths of finite $p$-variation for $p \geq 1$, see, *e.g.*,  *Lyons 2014* and many references cited therein."
>
> - **Matrix Exponential**
>
> To efficiently evaluate and train the development layer, we include the PyTorch implementation of the matrix exponential and its differential at the end of Section 3 of our manuscript; see also the ExpRNN paper, available at   https://github.com/Lezcano/expRNN. We refer the reviewer to [4,5] for in-depth discussions on efficient computation techniques for the matrix exponential. In numerical experiments, we used the moderate matrix order ($m \leq 30$) to achieve satisfactory performance, with the corresponding training time standing at a manageable level (see Training time \& Memory in Section 4.1). Though beyond the scope of our paper, we concur with the reviewer that the matrix exponential computation might become a bottleneck when using a very large matrix order, which certainly deserves further investigation.
>
> - **Universality and characteristicity:**
>
> In our paper (as with many other references),  the characteristic property means that the path development uniquely determines the path, which is equivalent to that the linear maps $M$ in Theorem 2.1 separate points over the space of signatures. See Theorem 2.1 and Remark 2.3.
> In the revised paper, we added the reference Theorem B.1 and Theorem 2.1 immediately after " characteristic and universal" in the first point of Key advantages for enhanced clarity.
>
> - **Run time and model complexity**:
>
> We concur with the reviewer on the importance of including discussions on computational complexity. Indeed,  the subsection ``Training time \& Memory'' of Section 4.1 in the earlier submission contains a comparative study on the numerical results between the path development and other models. To reinforce this point, at the end of Section 3 in the revised version we explicitly listed the computation complexity of our development layer (Algorithms 1 and 2). In particular, the computation time and storage are *linear* in the time dimension of the path. See also our response to "matrix exponential" for the complexity of matrix exponential computations.

---

> ### Author Response · Authors · 2024-06-27
>
> Reply to Questions section:
>
> - Spatial data: Originated from the solution to the controlled differential equation driven by a path $X: [0, T] \rightarrow \mathbb{R}^{d}$, our development layer is designed specifically for time series data rather than spatial data. The notation at the end of p.2 represents $\mathbb{R}^d$-valued time series. Meanwhile, as mentioned in the last sentence of the conclusion section, one may further extend the path development layer to handle spatio-temporal data; *e.g.*, rigid body and temporal graph.
>
> Reply to Minor section:
>
> - We deleted "will" in this sentence.
>
> - We corrected the typo by changing "on-Euclidean" to "non-Euclidean".
>
> - We adjusted the fonts in Table 1.
>
> References:
>
> [1] Stable Architectures for Deep Neural Networks, Inverse problems, 34(1), p.014004.
>
> [2] Anti-Symmetric DGN: a stable architecture for Deep Graph Networks.  In The Eleventh International Conference on Learning Representations.
>
> [3] Feature Transportation Improves Graph Neural Networks, In Proceedings of the AAAI Conference on Artificial Intelligence (Vol. 38, No. 11, pp. 11874-11882), 2024.
>
> [4] Awad H Al-Mohy and Nicholas J Higham. Computing the Frechet derivative of the matrix exponential, with an application to condition number estimation. SIAM Journal on Matrix Analysis and Applications, 30(4):1639–1657, 2009.
>
> [5] Awad H Al-Mohy and Nicholas J Higham. A new scaling and squaring
> algorithm for the matrix exponential. SIAM Journal on Matrix Analysis and
> Applications, 31(3):970–989, 2010

---

> ### Comment · Reviewer_cmMa · 2024-06-29
> **Thanks for your revision**
>
> I appreciate the authors' revision and responses which have clarified many of my questions. Also, apologies for not noting the reported runtime in your original submission. I think that discussing the computation of matrix exponentials as you did in the revision is valuable for the community, so thank you.
>
> I only have a comment about the proposed references [1-3], while they do not directly discuss time series data, my comment was that they are relevant because they view layers as time (as time steps of ODE integration). Furthermore, in [3] there is an element of time series data because it considers spatio-temporal data, so I believe there is a relation to your work.

---

> > ### Author Response · Authors · 2024-06-30
> >
> > We are pleased to hear that you are satisfied with our response and the revisions made to the paper. Thanks very much for your clarification about the link between our work with the references [1-3]. To address your comment, we have added [1-3] to Section 1.1, Related Work/Continuous Time Series Modelling. Additionally, the discussion on Geometric Deep Learning addresses the relationship between path development and ResNet, which has a similar analogy to Neural ODE that views layers as time. For reference [3], we have added a new Section 5.3, Hybrid Model Architecture, to discuss future work on path development that can effectively model more complex data streams, including spatio-temporal data.
> >
> > References:
> >
> > [1] Stable Architectures for Deep Neural Networks, Inverse problems, 34(1), p.014004.
> >
> > [2] Anti-Symmetric DGN: a stable architecture for Deep Graph Networks.  In The Eleventh International Conference on Learning Representations.
> >
> > [3] Feature Transportation Improves Graph Neural Networks, In Proceedings of the AAAI Conference on Artificial Intelligence (Vol. 38, No. 11, pp. 11874-11882), 2024.

---

> > > ### Comment · Reviewer_cmMa · 2024-07-01
> > > **Thank you for the response**
> > >
> > > I thank the authors for their response and revisions.

---

> > > > ### Author Response · Authors · 2024-07-01
> > > >
> > > > You are welcome. Thank you once again for your constructive feedback.

---

### Review · Reviewer_Kb6t · 2024-06-20

**Summary Of Contributions:**

The authors propose a novel neural network architecture (via a novel layer) for
time series data, called the path development layer, which is derived from the
notion of the development of a path in rough path theory. This formalism
essentially corresponds to a generating function for the signature of a path on
$\mathbb{R}^d$ -- a useful feature representation for suitably-encoded time
series data, but which is not very scalable -- with monomials in
a sub-Lie-algebra of $\mathfrak{gl}(n)$. The idea is then to consider
a discretization of an arbitrary path into linear segments, in which case the
dynamics associated to the path's development have a simple closed-form
expression in terms of the matrix exponential, and interpret the resulting
discrete iteration as a neural network (composed of the so-called path
development layers). As a result, the dimensionality of this network depends
only on $n$, not $d$; this can lead to significant savings over calculating the
naive signature, without sacrificing too much 'capacity' of the underlying
representation.

The authors' contributions are to clearly derive this formalism from basic
principles in rough path theory, then describe a computational implementation
for it using ideas from Riemannian optimization (this is used for learning the
'parameters' of the path development layer, which correspond to matrix-valued
maps from tangent vectors along the discretized path to elements of the Lie
algebra). They demonstrate the resulting layer numerically on several standard
benchmarks from this area, showing that it leads to improved performance and
scalability versus previous deep approaches for approximating the signature,
as well as competitive performance with black-box approaches when used in
combination with a LSTM.

**Audience:**

Yes

**Claims And Evidence:**

Yes

**Requested Changes:**

- Discussion of similar Riemannian-inspired NN architectures mentioned above
- If possible, a more formal example of the tradeoffs between $m$/number of
  developments in a specific toy case that gives more precise insight than Table
  1/Figure 3 would be very helpful

**Strengths And Weaknesses:**

## Strengths

- The authors' method is principled, and the paper presents a very careful and
  complete derivation of the method from basic principles from rough path
  theory and controlled differential equations theory. The mathematical writing
  is clear and precise, and examples and remarks given along the way make this
  material to a technical but out-of-area reader. Complete proofs of claims are
  provided in appendices. These derivations actually seem to represent a useful
  introduction to some of these concepts (e.g. the signature) versus other more
  technical papers in the literature.
- The authors demonstrate their method on a range of experiments, from standard
  time series benchmarks (e.g., audio data with cepstrum features) to dynamics.
  The method effectively improves performance and scalability with regards to

## Weaknesses

- The numerical performance of the authors' method is often significantly below
  that of pure black-box deep learning approaches (e.g., LSTM or SSMs), whereas
  a 'hybrid' approach involving their layer (playing the role of a kind of
  'feature extractor') can often improve over this baseline. The authors mention
  that their method is more robust and more easily trainable, as well, but
  I have some doubts about this type of claim: it seems more likely to me (e.g.,
  some evidence suggested by experiences with past networks with a similar
  flavor in the image/CNN context, such as the scattering networks) that any
  benefits along these two axes are due to inherent constraints on the capacity
  of networks derived from the authors' layer, and therefore at odds with
  performance (which is not the tradeoff one desires).

- The authors' layer is presented as a complete novelty, but it shares some
  functional similarities with other types of layers that have been inspired by
  Riemannian optimization: for example, the Riemannian ResNet (https://arxiv.org/abs/2310.10013) proposes to use
  the exponential map similar to a skip connection. One major difference here
  appears to be the focus on time series data and the principled foundation in
  terms of the signature, but the authors should compare to these prior works in
  order to situate their network architecture contribution within the deep
  learning literature.

## Detailed Comments

- Theorem 2.1: $x$ is defined with subscripts but the hypothesis involves $x$
  with a superscript (is it supposed to be the same?). The authors should
  comment here on how this dimension reduction functions in practice, given that
  the theorem concerns a noncompact Lie group and an earlier remark in the paper
  stated that noncompactness could be a problem in practice; can a result like
  Theorem 2.1 hold for a compact Lie subalgebra as well? It would be most useful
  to have a more "quantitative" version of this result as well (e.g., is it
  possible to assert 'how well' I can approximate a path using $k$ terms of
  the signature sequence and $m \times m$ operators, for any $k, m$?)
- Claim before Remark 3.1: the use of the product shorthand notation here
  suggests that these matrix exponentials are commuting; but isn't it the case
  that they only commute when $M_{\theta}$ has a far more restrictive structure?
- Table 1 and Figure 3 give a good empirical sense of the role of the choice of
  operator dimension and number of developments in determining performance of
  the authors' method. However, given the very significant mathematical flavor of
  the paper, it feels like there is a missed opportunity to demonstrate some toy
  examples in which this tradeoff can be mathematically-precisely understood. Are there
  natural simple curves (e.g. like in the toy Figure 2 example, which does not
  really play a role in the current text) where the kind of 'low dimensional
  structure' inherent in the signature is present and moreover captured by the
  authors' proposed layer at certain choices of $m$? This would also go a long
  way to demonstrating what the limitations of the authors' approach might be.

## Minor Comments

- The beginning of the intro feels slightly inaccessible to a non-specialist in
  rough path theory. Two points that feel improvable: (i) in the sentence
  "roughly speaking" that describes the signature, the description seems both
  overly formal and insufficiently precise (the formalism presented is a bit
  unwieldy with notation that can't be understood based on the description
  given; it also suppresses/does not define the "iterated integral"
  notation appearing here), maybe prefer to add a pointer to appendix A here
  and add clarification of iterated integral notation to appendix A; (ii)
  although it will be spun out in the remainder of the paper, it would be
  helpful to strengthen the transition from the mathematical idea of the
  signature to the idea of the signature as a feature map by explaining more
  clearly how one computes with/approximates the signature (the current 'roughly
  speaking' formal description does not appear to admit an easy efficient
  approximation).
- The paper's primary focus and contributions are around proposing a new
  computational method for computing with time series data which is a scalable
  alternative to the signature -- hence practical -- but the paper is written in
  an overly-mathematical style that at times introduces too much distracting,
  irrelevant detail with regards to this goal. For example, it seems like
  section 3.2 could be replaced by a reference to the standard Absil, Mahoney,
  Sepulchre and maybe Remark 3.3 kept (because this is simply a standard
  implementation detail, which the matrix exponential implementation should
  handle regardless). On the other hand, this style of writing is helpful in
  sections that are closer to the conceptual novelty of the authors' approach
  (e.g. section 2).

## Typos/etc.

- Page 2 second graf: "will shall" (sic)
- display on page 2: what is the 0,T subscript's meaning (is this a typo and it
  should be $[0, T]$)?
- Equation (4) and Lemma 2.2 - there might be a 'types' issue here? ($\tilde{M}$
  is defined on infinite sequences of tensors of increasing "order"
  $T(\mathbb{R}^d)$, but (4) only shows how $\tilde{M}$ acts on a single element
  of this infinite sequence, and for 2.2 to be true, I think the expression in
  (4) needs to be summed over "order" (as in the display on page 2?).
- Lemma 2.3 - is there a typo in the conclusion? ($X$ is defined as a path on
  $[0, T]$; what is the meaning of the double subscript $X_{s, t}$ (etc.)?)
- Lemma 2.4 - please correct me if I'm mistaken, but this seems wrong by
  pattern-matching against Theorem 2.2, first result (e.g., the chain rule in
  euclidean space gives that the gradient of x \mapsto f(A(x)) for a linear
  map A is A^* \nabla f(A(x)) (^* denotes adjoint); so why is there no Proj
  operation "on the inside" of the composition here?). I am also not sure why
  the differential notation is different here than in Theorem 2.2 (why d_{M^T}
  instead of df_{M^T}?).
- page 13: broken label (latex issue) for Figure 3 (the referrer is incorrectly
  written "Section 4.1.1")

---

> ### Author Response · Authors · 2024-06-27
>
> Reply to Weakness section:
>
> - 1. **Comparison with Black-box Deep Learning Approaches**: (a) You correctly note that the path development alone may underperform the black-box neural networks (*e.g.*, LSTM), while the hybrid model with development shows superior performance over the baseline. (b) Typically, the feature extractor is applied to raw data before the black-box models. However, in this hybrid model, the development layer follows the LSTM, and hence it does not play a role as the feature extractor. We believe that the improvement over LSTMs stems from our use of SO-groups based development layer, which alleviates the gradient issues. See Remark 3.4 for details. (c) Note that the utilization of the path development can extend beyond the current hybrid LSTM+Dev model. As pointed out by the reviewer, the path development can be used as a *feature extractor* of time series, combined with other general neural networks. The implementation of the path development layer allows it to be conveniently applied to general time series data; meanwhile, with the appropriate choices of neural networks which follow this layer, the resulting hybrid models may further enhance the baselines.
>
>  2. **Claims of Robustness & Trainability**: We claimed that the path development layer improves the robustness of LSTM (see Table~3), possibly due to the continuous nature of the development. We also claimed that the hybrid model improves the training stability of  LSTM and reduces the need for hyper-parameter tuning (see Figure 3). Moreover, the competitive performance with better efficiency of the hybrid models is highlighted --- they achieve significantly better performance than the continuous time series model ($5\times$ faster training time; Section 4.1, Training time & Memory), and achieve results comparable to large-scale state-space models (using $\sim 1/4$ number of parameters).
> - Thank you for pointing out the reference on Riemannian ResNet, which is now included in the related work section. We clarified its connections with and differences from our path development approach. As you correctly noted, Riemannian ResNet employs the exponential map similarly to our path development layer. However, the recurrence of Riemannian ResNet lies in two consecutive layers ($t$ = the depth of layers), while for us the recurrence occurs between two consecutive times in a single layer. Meanwhile, our development layer is designed specifically for times series input, while the Riemannian ResNet is not.
>
> Reply to Detailed Comments:
>
> - We corrected the typo by changing the superscript of $x$ to the subscript.
>
> - Theorem 2.1 (the characteristic property of development) ascertains that symplectic algebra $\mathfrak{sp}(m,\mathbb{C})$-valued linear maps separate points over the signatures. In fact, for any Lie algebra (e.g., $\mathfrak{gl}(m,\mathbb{C})$), which contains $\mathfrak{sp}(m,\mathbb{C})$, the corresponding development satisfies the characteristic property. In this regard, whether the corresponding Lie group ${\rm SP}(m,\mathbb{C})$ is compact plays no role. However, as pointed out in Remark 3.4, the gradient issues associated with RNNs are alleviated when the Lie group contains \emph{isometries}. Thus we consider instead ${\rm SP}(m,\mathbb{C})\cap U(2m,\mathbb{C}) =: {\rm SP}(2m,\mathbb{R})$, the so-called \emph{compact symplectic group}, as in Remark 2.1.
>
> - We concur with the reviewer that it is important to prove quantitative results on the approximation of paths via the signature or path development. However, to our best knowledge, this seems elusive in the existing literature.
> - The shorthand $\prod$ for the product in the sentence above Remark 3.1 was changed to
> $z_n = \exp(M_{\theta}(x_1-x_0)) \cdot \exp(M_{\theta}(x_2-x_1))\cdots \cdot \exp(M_{\theta}(x_{n}-x_{n-1}))$, so as to emphasise the non-commutative nature of the product here. %The particular order of matrices in the product above cannot be altered.
>
> - We frankly admit that it is challenging to provide informative toy examples illustrating how the ``low-dimensional structures'' inherent to the signature are captured. However, we highlight Remark 2.3 as a qualitative statement of the dimension reduction benefit offered by the development (a cross-validation of [1, Theorem~2.1]). We also point out the recent work [2], in which an explicit, rigorous algorithm is presented for recovering the $k^{\text{th}}$ level of the signature using only the unitary development into $(k+1) \times (k+1)$ matrices.

---

> ### Author Response · Authors · 2024-06-27
>
> Reply to Minor Comments section:
>
> - We modified the beginning of the introduction according to your suggestions.
>
> (i) We changed the sentences "Roughly speaking ... many references cited therein" to the following:
>
> >Roughly speaking, the signature of a path serves as a principled feature, offering a top-down description of the path. Just as the monomial basis of $\mathbb{R}^d$, the signature --- viewed as their noncommutative analogue --- constitutes a basis for the path space. More specifically, the signature of $X$ is defined as an infinite sequence  $(1, X_{[0,T]}^{(1)}, \cdots,  X_{[0,T]}^{(k)}, \cdots )$, where $$X_{[0, T]}^{k} = \int_{0 < t_{1} < \cdots t_k<  T} d X_{t_1} \otimes \cdots \otimes d X_{t_k},$$ provided that the above integral is well defined. We refer the reader to Section~2.1 and Appendix A for the precise definition of the signature of paths of bounded variation. For a more general case of paths of finite $p$-variation, $p \geq 1$, see, *e.g.*, Lyons 2014 and many references cited therein.
>
> (ii) To further illustrate our changes in (i) above, we added the explicit formulae for the signatures of linear and piecewise linear paths in Appendix A, followed by a remark on how to extend these examples to general continuous paths with references. We hope the reader may find it helpful for the computation of signature of the times series data.
>
> - We have simplified the mathematical presentations in Section 3. For example, detailed explanations for metrics on the matrix Lie group, originally located between Remark 3.3 and Proposition 3.1, have been moved to the appendix.
>
> Reply to Typos/etc section:
>
> - We delete "will".
>
> - It should be $[0, T]$.
>
> - Equation (4) and Lemma 2.2: Note the $\widetilde{M}$ is defined as a linear map from $T((\mathbb{R}^d))$ to $\mathfrak{gl}(m, \mathbb{F})$, hence is determined by its values on the canonical basis of $T((\mathbb{R}^d))$, namely that $\{ e_{i_1} \otimes e_{i_2} \otimes \cdots \otimes e_{i_n}:n \in \mathbb{N};\, i_1, \cdots, i_n \in \{1, \cdots, d\}\}$.
>
> - Lemma 2.3: Yes, this is a typo. We changed the subscript $s, t$ ($\lambda_s, \lambda_t$, resp.) to $[s, t]$ ($[\lambda_s, \lambda_t])$, resp.);  here $X_{[s, t]}$ is $X$ restricted to the interval $[s, t]$.
>
> - Lemma 2.4: This is a typo. Indeed, $Proj$ is missing inside the composition of $f \circ \exp$ on the right-hand side of the equation. To be consistent with the notations throughout the proof, we revised  Lemma 2.4 as follows:
>
> $$
> \nabla (f \circ \exp \circ Proj)(A) =  Proj \\{ \{d_{M^T} \exp(\nabla f \circ \exp(M))\} \\},
> $$
>
> where $A\in \mathfrak{gl}(m;\mathbb{C})$ and  $M =  Proj(A)$.
>
> - page 13: We changed "Section 4.1.1" to "Figure 3".
>
>
> Reply to Requested Change section:
>
> - Discussions on the Riemannian-inspired neural network architecture are added to the related work section, as outlined in the second point of the "Reply to Weakness" section.
>
> - Please see the last point of our response to the Detailed Comments section.
>
> References
>
> [1] Chevyrev, Ilya, and Terry Lyons. "Characteristic functions of measures on geometric rough paths." (2016):  The Annals of Probability, 4049-4082.
>
> [2] On the determination of path signature from its unitary development. arXiv preprint arXiv:2404.18661. 2024.

---

> > ### Comment · Reviewer_Kb6t · 2024-06-28
> > **thanks**
> >
> > Dear authors,
> >
> > Thank you for the thorough response to my review, and for the many edits you have made to the paper. Thank you as well for correcting my misconceptions regarding the use of your method in hybrid architectures and other aspects of performance in the experiments. I am satisfied with the arguments you have made.

---

> > > ### Author Response · Authors · 2024-06-28
> > >
> > > We are pleased to hear that you are satisfied with our response and the revisions made to the paper.  Thank you once again for your valuable feedback and suggestions!

---

### Author Response · Authors · 2024-06-27
**Reply to all**

We thank all the reviewers for their insightful comments and constructive suggestions. We are pleased that all the reviewers find our proposed method conceptually novel, technically correct, and theoretically motivated. All reviewers also recognize that our proposed model addresses important problems in time series modelling and demonstrates its capability through a wide range of experiments. Questions and comments from each reviewer are addressed below in respective sections. Additionally, we uploaded the revised manuscript with the modifications highlighted in red. Please kindly let us know if there are any further questions we should address. Thank you very much.

---

### Decision · Action_Editor_WaXe · 2024-08-22

**Recommendation:** Accept as is

**Comment:**

The paper introduces a new layer for time series neural networks termed the path development layer, which leverages tools from rough path theory to provide more efficient estimates in the case of high-dimensional paths, along with suitable optimization procedures. The reviewers found the theory and experiments novel, interesting, and thorough, though the presentation is a little technical, and the empirical results slightly underwhelming.

**Audience:**

yes

**Claims And Evidence:**

yes